# CALM 🪶 Before the STORM ⚡ :
# Unlocking Native Reasoning for Optimization Modeling

**Zhengyang Tang** [* 1 2]  **Zihan Ye** [* 1]  **Chenyu Huang** [* 3]  **Xuhan Huang** [1]  **Chengpeng Li** [2]  **Sihang Li** [2]
**Guanhua Chen** [4]  **Ming Yan** [1]  **Zizhuo Wang** [1]  **Hongyuan Zha** [1]  **Dayiheng Liu** [2]  **Benyou Wang** [1 5]

## Abstract

Large Reasoning Models (LRMs) create new opportunities for automating optimization modeling, but they also make post-training more delicate. In this task, strong performance often requires the model to formulate the problem, write solver code, run it, inspect the output, and revise when needed. We show that directly fine-tuning LRMs on already written-out Operations Research (OR) solutions can improve easier cases while hurting harder ones, suggesting that this training signal can interfere with the model's own way of solving the task. We therefore propose **CALM** (*Corrective Adaptation with Lightweight Modification*), which lets the base LRM attempt the problem first, then inserts a short hint at the first detected mistake and lets the model continue from there. These hints modify fewer than 2.6% of generated tokens. The corrected solutions are used for supervised fine-tuning and then reinforcement learning, producing **STORM**, a 4B optimization-modeling specialist that reaches 68.9% macro-average accuracy across five benchmarks and matches 671B DeepSeek-R1-0528. Under a matched hard-benchmark control, CALM also yields stronger final RL performance than direct distillation baselines that train on complete teacher-generated solutions from much stronger models. Overall, for this task, local repair of the base model's own solution is more effective than full teacher-solution replacement. Code and models are available at https://github.com/tangzhy/STORM.

---

[*]Equal contribution  [1]The Chinese University of Hong Kong, Shenzhen, China  [2]Qwen Team, Alibaba Group, Beijing, China  [3]Shanghai University of Finance and Economics, Shanghai, China  [4]Southern University of Science and Technology, Shenzhen, China  [5]Shenzhen Loop Area Institute, Shenzhen, China. Correspondence to: Dayiheng Liu <liudayiheng.ldyh@alibaba-inc.com>, Benyou Wang <wangbenyou@cuhk.edu.cn>.

*Proceedings of the 43rd International Conference on Machine Learning*, Seoul, South Korea. PMLR 306, 2026. Copyright 2026 by the author(s).

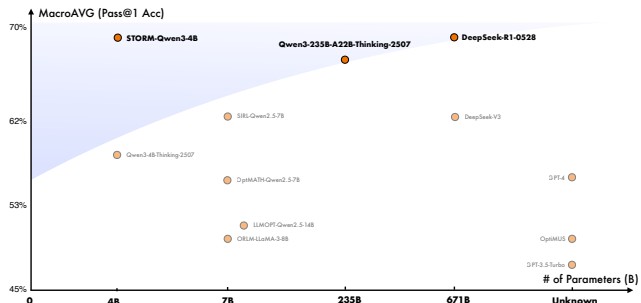

**Figure 1.** Performance comparison across different model sizes. The proposed STORM-4B matches the performance of models orders of magnitude larger (e.g., the 671B DeepSeek-R1-0528).

## 1. Introduction

Operations Research (OR) and optimization modeling are central to decision-making in areas such as inventory management and airline crew scheduling (Silver, 1981; Vance et al., 1997). Yet turning a real-world problem description into a correct mathematical program and executable solver code remains difficult and labor-intensive (Huang et al., 2025). This has made optimization modeling a natural target for language models. Recent systems show that LLMs can help formulate OR problems and write solver programs from natural language (Huang et al., 2025; Jiang et al., 2024; Chen et al., 2025). Still, this task is not just about writing a final script. Strong performance often requires the model to formulate the problem, write solver code, run it, inspect the output, and revise when the run reveals a modeling mistake, a coding bug, or an implausible answer.

The arrival of Large Reasoning Models (LRMs), such as Qwen3 and DeepSeek-R1, changes how this task should be adapted (Qwen Team, 2025; DeepSeek-AI, 2025). Unlike earlier instruction-tuned models, these systems reveal a much richer multi-step solution process before producing the final answer. Earlier OR training recipes were developed mainly for the older setting, where it was natural to train on collections of problems paired with completed solutions (Huang et al., 2025; Jiang et al., 2024). With LRMs, however, the base model already has its own way of working through the problem. The key question is therefore whether the standard OR supervision still helps in this set-

ting, or instead replaces the model's own solution process with externally provided patterns.

We test that standard recipe directly by fine-tuning Qwen3-4B-Thinking-2507 on OR-Instruct-3K (Huang et al., 2025). The result is mixed: performance improves on easier benchmarks but drops sharply on harder ones. This is the first sign that the old recipe does not transfer cleanly to LRMs. For these models, directly imitating already written-out solutions may help on short and routine cases, but interfere with the model's own multi-step solution process when the task becomes harder.

This result points to a more concrete question: if training on completed solutions is unreliable for LRMs, where does the base model's own solution process actually break down? To answer this, we analyze the mistakes made by the base LRM on optimization problems. We find seven recurring error types, which fall into two broader groups. The first is *Code Utilization Distrust*: the model avoids or underuses the solver, for example by doing numerical work in natural language, splitting code into unhelpful fragments, or rechecking values the solver has already returned. The second is *Lack of OR Expertise*: the model makes formulation, constraint, or implementation mistakes that reflect missing domain knowledge. These patterns suggest a simple design principle: instead of rewriting the whole solution, training should identify the first concrete mistake, correct that point, and allow the model to continue its own reasoning process.

We therefore propose **CALM** (*Corrective Adaptation with Lightweight Modification*), a framework built around this principle. CALM first lets the base model attempt the problem on its own. An intervener then checks the current solution, identifies the first mistake, inserts a short hint at that point, and lets the model continue from the revised context. The intervention is deliberately small: across the curated data, CALM changes fewer than 2.6% of generated tokens, leaving the rest of the model's solution intact. We keep only those corrected solutions that are both numerically correct and judged to follow a clean solving process. These examples are then used first for supervised fine-tuning and then for reinforcement learning. The final model is **STORM** (*Smart Thinking Optimization Reasoning Model*).

Across five optimization modeling benchmarks, STORM, built from a 4B LRM, reaches 68.9% macro-average accuracy and matches the performance of 671B DeepSeek-R1-0528. Later experiments sharpen the main claim. Under a matched hard-benchmark control, even direct distillation baselines—which fine-tune on complete teacher-generated solutions from much stronger models and then apply the same downstream RL recipe—remain worse than CALM. Swapping Gemini-2.5-Pro for open-source DeepSeek-R1-0528 as the intervener changes the final five-benchmark macro average only from 68.9% to 67.8%. Taken together,

these results support a narrow message: on this task, it is better to keep most of the base model's own solution and repair it locally than to replace it with a complete teacher-generated solution.

Our contributions are as follows:

- We show that the standard OR training recipe used for earlier instruction-tuned models does not transfer cleanly to LRMs: direct fine-tuning on OR-Instruct-3K helps easier benchmarks but hurts harder ones.

- We identify two recurring sources of failure—*Code Utilization Distrust* and *Lack of OR Expertise*—and propose **CALM**, which fixes the first concrete mistake with a short hint instead of rewriting the whole solution. In our curated data, CALM changes fewer than 2.6% of tokens.

- We build **STORM**, a 4B optimization-modeling specialist that reaches 68.9% macro-average accuracy across five benchmarks and matches the performance of 671B DeepSeek-R1-0528.

- We provide controlled evidence that sharpens the paper's claim: under a matched hard-benchmark control, CALM gives a better starting point for downstream RL than direct distillation baselines and remains robust to the choice of intervener.

We situate our work within the broader literature and provide a discussion of related work in Appendix A.

**Conflict of Interest Disclosure.** Authors Z.T., C.L., S.L., and D.L. are employed by Alibaba Group, which leads the development of the Qwen model family. Qwen3 models were used as the base model and among the baselines evaluated in this paper.

## 2. Background and Motivation

### 2.1. Background: Solving Optimization Problems with LLMs

Automated optimization modeling is the task of translating a natural language problem description into a mathematical model and executable solver code (see Figure 2). For evaluation, the solver computes a candidate solution, which is deemed correct if its objective value lies within a predefined relative error of the ground truth. Performance is assessed on benchmarks that span a range of difficulty, from easier problems in `NL4Opt` to complex industrial cases in `IndustryOR`. A detailed overview of these benchmarks is provided in Appendix E.1.

As shown in Figure 2, there are two simple ways to think about this task.

**Write the whole solution in one pass.** Earlier systems often read the problem description and then directly write

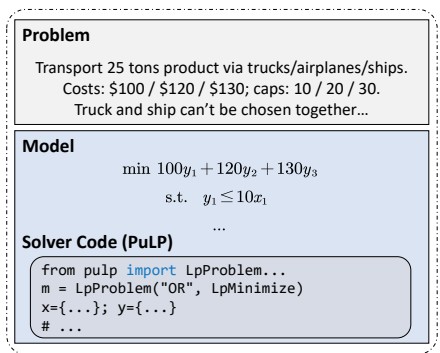

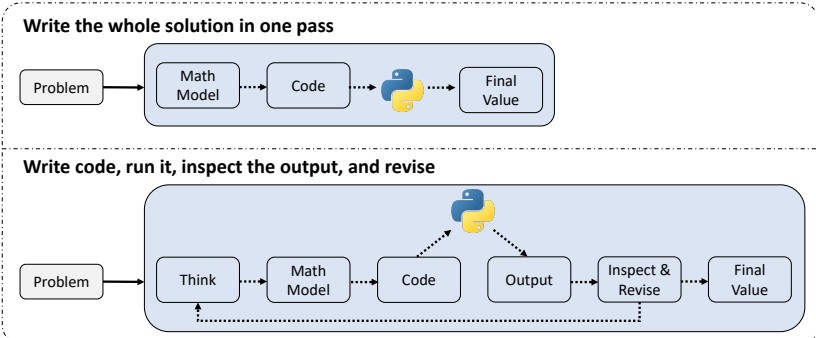

**Figure 2. Left:** An example showing the translation from a natural language problem to mathematical models and solver code. **Right:** Two ways to solve the task: write the full model and code in one shot (top), or write code, run it, inspect the output, and revise when needed (bottom).

the mathematical model and solver code in one shot (Huang et al., 2025; Jiang et al., 2024). This is a natural training recipe when the supervision is also given as completed solutions.

**Write code, run it, inspect the output, and revise.** With modern LRMs (Qwen Team, 2025; DeepSeek-AI, 2025), the base model often produces long intermediate reasoning before the final answer. In optimization modeling, this makes it natural to let the model write solver code, run it, inspect the output, and revise when the run reveals a modeling mistake, a coding bug, or an implausible answer. The key question for adaptation is then not only what final answer to supervise, but how to improve this process without replacing it with another complete solution written elsewhere.

### 2.2. Pilot Study: Fine-Tuning on OR-Instruct-3K

Given the availability of open-source LRMs and OR-Instruct-3K (Huang et al., 2025), the most direct adaptation strategy is to fine-tune an LRM on these already written complete solutions. We use this as a pilot study to test whether the standard recipe from earlier instruction-tuned models still works once the base model is an LRM.

**Table 1.** Performance of the base LRM and the same model after fine-tuning on OR-Instruct-3K. Gains on easier benchmarks come with large drops on harder ones.

| Benchmark | Base LRM | + SFT on OR-Instruct-3K | Absolute Change |
|---|---|---|---|
| NL4OPT | 85.8 | 92.9 | +7.1 |
| MAMO Easy | 73.8 | 88.7 | +14.9 |
| MAMO Complex | 46.5 | 40.5 | -6.0 |
| IndustryOR | 46.2 | 27.5 | -18.7 |
| OptMath | 33.1 | 6.6 | -26.5 |
| **Macro AVG** | **57.1** | **51.2** | **-5.9** |

The results in Table 1 show a clear trade-off. Fine-tuning on OR-Instruct-3K improves the easier benchmarks, especially `NL4Opt` and `MAMO-Easy`, but sharply degrades the harder ones, especially `IndustryOR` and `OptMath`. The full experimental setup is described in Appendix E.2.

A simple explanation is that OR-Instruct-3K asks the model to imitate a completed solution. On easier problems this can help. On harder problems, however, it can pull the LRM away from its own write–run–inspect–revise behavior and toward a more rigid one-shot writing style (Zhang et al., 2025). This is the mismatch that motivates our method. Later experiments revisit this question under stronger controls. Here, the pilot study serves only to show that the old recipe no longer transfers cleanly to LRMs.

### 2.3. A Taxonomy of Failures in the Base LRM's Solutions

Our pilot study shows that simply fine-tuning on completed solutions is unreliable. The next question is where the base model's own solution process breaks down. If we want to keep most of that process and only fix the bad parts, we first need to know what those bad parts are.

**Establishing a Protocol for Flaw Identification.** To perform a rigorous analysis, we establish a systematic protocol. We first prompted a base LRM to generate solutions for a diverse set of problems. A team of human experts with backgrounds in OR then analyzed these responses to identify recurring error patterns. Through a collaborative, multi-stage process of annotation, clustering, and refinement, the team converged on a set of seven distinct flaw types, which form the basis of our taxonomy. The complete, detailed protocol for this human-in-the-loop analysis is provided in Appendix C.

**A Two-Group View of the Errors.** Our analysis of the 7 identified error types shows that 6 are major errors that directly affect the solving process. The seventh is a minor formatting issue and is described in Appendix D. We focus here on the 6 major errors, which fall into two broader groups:

- **Code Utilization Distrust:** The model does extra work in prose instead of trusting the solver to do the computation. Typical cases include solving numerically in natural language, breaking the code into unhelpful frag-

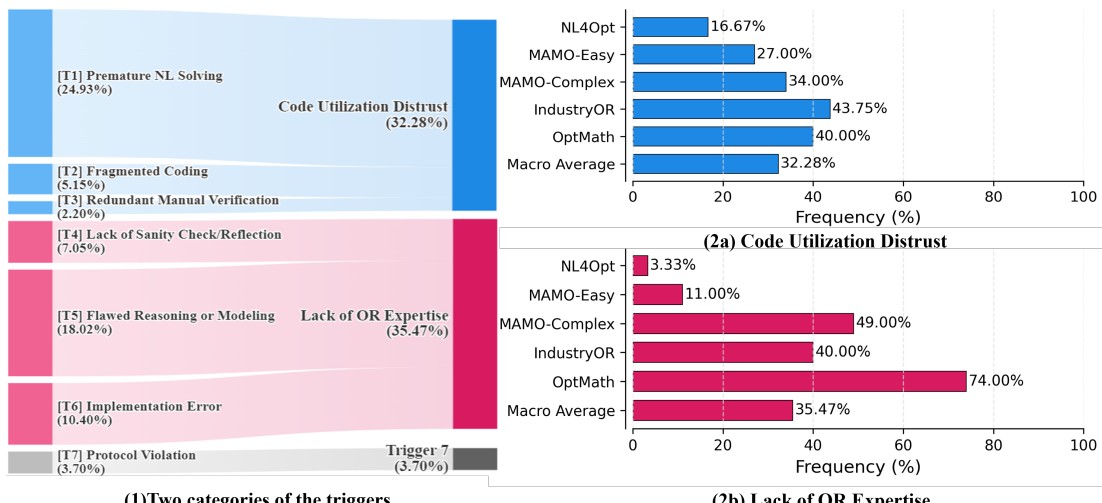

**Figure 3.** Common mistakes made by the base LRM. Left: the average frequency of each mistake type. Right: how the two main groups of mistakes vary across benchmarks.

ments, or manually re-checking values that the solver has already returned.

- **Lack of OR Expertise:** The model makes formulation, constraint, or implementation mistakes that reflect missing domain knowledge. Typical cases include wrong mathematical formulations, missed constraints, and coding errors that do not faithfully implement the intended model.

This grouping gives us a concrete picture of what needs to be fixed during adaptation.

**Quantifying Error Patterns across Benchmarks.** With this taxonomy in place, we quantify these errors at scale using an expert-level LLM as a consistent annotator. Details are provided in Appendix J. Figure 3 reveals a clear pattern: the main bottleneck changes with problem difficulty. On easier benchmarks such as NL4Opt and MAMO-Easy, the base model more often fails by not using the solver well. On harder benchmarks such as OptMath, the main problem is missing OR knowledge. Any successful adaptation therefore needs to do two things: help the model trust solver-based computation when it is appropriate, and correct the places where its OR modeling knowledge is still weak.

## 3. Methodology

### 3.1. Preliminaries: Formalizing the Base Model's Solution Trajectory

We model the base LRM's attempt to solve a problem as a trajectory in a code-execution environment $E$. Given a problem $P$, the model alternates between writing text reasoning $s_t$ and emitting a code block $a_t$. The code is executed by $E$, which returns an output $o_t$. The full trajectory after $T$ steps is

$$\tau^{(T)} = (s_0, a_0, o_0, s_1, a_1, o_1 \ldots, s_T, a_T, o_T), \quad (1)$$

where $s_t$ and $a_t$ denote the text and code at step $t$, and $o_t$ is the execution result returned by the environment. The trajectory evolves as

$$(s_t, a_t) = \pi_\theta(\tau^{(t-1)}), \quad o_t = E(a_t),$$
$$\tau^{(t)} = \tau^{(t-1)} \oplus s_t \oplus a_t \oplus o_t. \quad (2)$$

Our goal is to train $\pi_\theta$ so that these trajectories end in correct solutions.

### 3.2. CALM: Correcting the First Mistake

CALM is built around a simple idea: keep most of the base model's own solution, and intervene only where it first goes wrong. Instead of replacing the whole solution with one written elsewhere, CALM inserts a short corrective hint at the first concrete mistake and then lets the base model continue. We refer to the base model as the *Reasoner* and to the hint-giving model as the *Intervener*.

**Targeted Hints for Common Errors.** CALM uses the error types in Section 2.3 to decide what kind of hint to add (see Appendix D). Two cases are especially important:

- *For Code Utilization Distrust*: If the model starts solving numerically in prose or keeps breaking the code into small unusable pieces, the hint pushes it back toward solver-based computation. For example, once the formulation is already available, the hint can tell the model to write one complete `pulp` program and let the solver compute the answer.

- *For Lack of OR Expertise*: If the model misses a key modeling concept, the hint points out the concrete issue that is breaking the solution. For example, if the model produces a fractional number of cars, the hint can remind it that the decision variables should be integers.

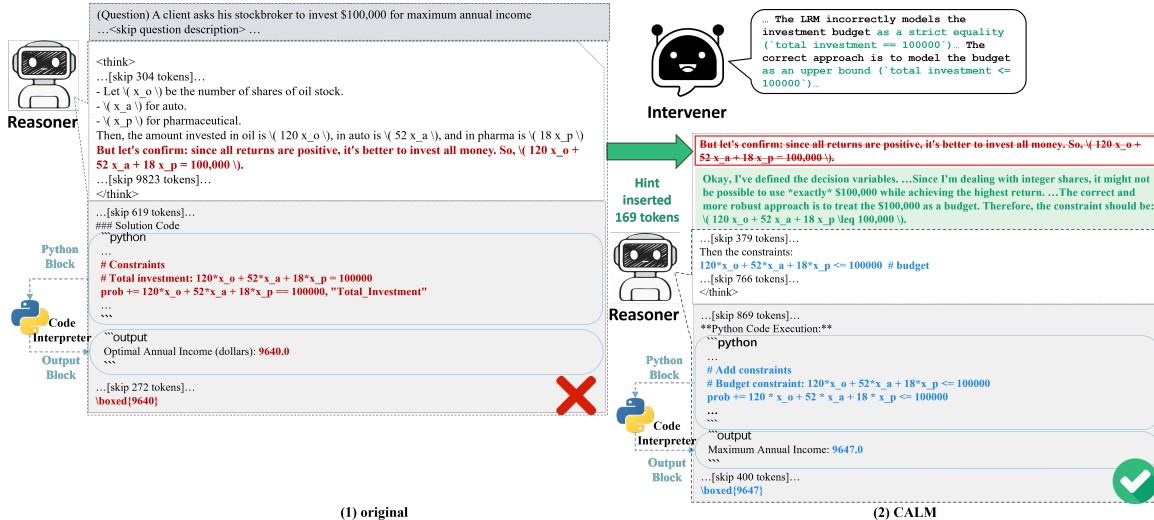

**Figure 4.** A representative *Lack of OR Expertise* error. **Left:** the base model writes an incorrect formulation and gets the wrong answer. **Right:** after one local hint, the model corrects the formulation and reaches the correct answer.

**The Local Repair Loop.** CALM turns an incorrect or low-quality solution into a better one through a local repair loop. Let $\tau^{(i)}$ denote the current solution at iteration $i$. The process proceeds as follows:

- **Initial Attempt ($i = 0$):** The Reasoner generates an initial solution $\tau^{(0)}$ for a given problem $P$.

- **Check the Current Solution:** The Intervener examines $\tau^{(i)}$. If no major mistake is found, CALM stops and keeps the current solution.

- **Fix the First Mistake:** Otherwise, the Intervener finds the first concrete mistake, inserts a short hint at that point, and lets the Reasoner continue from the revised context, producing $\tau^{(i+1)}$.

We cap the number of interventions to avoid unproductive repair cycles. The key property of CALM is locality: each intervention edits one concrete point instead of rewriting the whole solution. Appendix H provides more single-step examples, and Appendix I shows a complete multi-turn case.

**Filtering Corrected Solutions.** To ensure supervision quality, we build the SFT dataset $\mathcal{D}_{CALM}$ by keeping only corrected solutions whose final answer is correct and whose solving process is judged clean enough to train on (Figure 6).

### 3.3. Training Pipeline: CALM-SFT and RL

The corrected solutions produced by CALM are used in two training stages.

**Stage 1: Supervised Fine-Tuning.** We fine-tune the base LRM on $\mathcal{D}_{CALM}$ with a standard cross-entropy loss. The purpose of this stage is not to teach the model a completely new way of solving the task. Instead, it nudges the model toward better habits while still keeping the overall shape of its own solutions. Unlike a direct distillation baseline, the training target here is the base model's own solution after a small local repair.

**Stage 2: Reinforcement Learning.** We then continue with reinforcement learning so that the model can further improve correctness through its own exploration. We use Group Relative Policy Optimization (GRPO) (Shao et al., 2024), allowing interaction with the code interpreter for up to $T = 4$ code executions per rollout. The RL stage aims to maximize the expected reward $J(\theta) = \mathbb{E}_{\tau \sim \pi_\theta(\cdot|P)}[R(\tau)]$. Our reward is a simple binary signal based on the final outcome:

$$R(\tau) = \begin{cases} 1 & \text{if } \left| \frac{Ans(\tau) - Ans^*}{Ans^*} \right| \leq \epsilon, \\ 0 & \text{otherwise.} \end{cases} \quad (3)$$

where $Ans(\tau)$ is the final answer extracted from trajectory $\tau$, and $Ans^*$ is the ground-truth solution. We also apply execution-output masking during gradient computation to improve training stability. The final model is referred to as **STORM**.

## 4. Experiments

Our experimental evaluation first benchmarks **STORM** against leading models. We then ask what kind of supervision produces the gain, and test the main claim under matched controls.

### 4.1. Experimental Setup

**Benchmarks and Datasets.** Our evaluation is conducted on a diverse suite of five benchmarks: NL4Opt (Ramamonjison et al., 2023), MAMO-Easy, MAMO-Complex (Huang et al., 2024), IndustryOR (Huang et al., 2025), and OptMath (Lu et al., 2025). This selection, consistent with prior state-of-the-art studies (Chen et al., 2025), allows us

to rigorously test LRM capabilities across a spectrum of difficulty. All training and test data originate from a larger collection of public datasets (Jiang et al., 2024), which we have rigorously partitioned into non-overlapping training and test sets. A comprehensive breakdown of all data sources and our splitting strategy is provided in Appendix E.1.

**Baselines.** We benchmark STORM against a comprehensive set of baselines for a holistic performance evaluation. The comparison includes: (1) **Foundation Models**: GPT-3.5-Turbo, GPT-4 (Achiam et al., 2023) and DeepSeek-V3; (2) **Large Reasoning Models**: DeepSeek-R1-0528 (DeepSeek-AI, 2025) and Qwen3-235B-A22B-Thinking-2507 (Qwen Team, 2025); (3) **Agent-Based Methods**: Chain-of-Experts (Xiao et al., 2023) and OptiMUS (AhmadiTeshnizi et al., 2024); (4) **Learning-Based Methods**: ORLM (Huang et al., 2025), LLMOPT (Jiang et al., 2024), OptMath (Lu et al., 2025) and SIRL (Chen et al., 2025); and (5) crucially, our **Base LRM**, Qwen3-4B-Thinking-2507, which serves as the starting point to directly measure our framework's impact.

**Evaluation Protocol.** We report **pass@1** accuracy as the primary evaluation metric. To address the high variance of greedy decoding in LRMs, as noted in DeepSeek-R1 (DeepSeek-AI, 2025), we follow their recommended evaluation protocol. Specifically, for each problem, we generate 8 independent samples using their specified configuration (temperature=0.6, top-p=0.95). The final **pass@1** score is then reported as the average success rate across these 8 samples. This established method ensures a more robust and reproducible measure of a model's performance. For a fair comparison, all LRM-based models are evaluated under this protocol, allowing a maximum of 4 code executions per reasoning trajectory.

**Training Procedure.** For CALM data synthesis, we use Qwen3-4B-Thinking-2507 (Qwen Team, 2025) as the Reasoner and Gemini-2.5-Pro (Comanici et al., 2025) as the Intervener. The curated trajectories are then used in a two-stage training pipeline described in Section 3.3. Our final model, **STORM**, is obtained through this pipeline. Detailed implementations are provided in Appendix E.3.

**Matched Control Experiments.** In addition to the standard baselines above, we include several matched controls to test the paper's central claim. First, we compare different SFT targets on the same Qwen3-4B-Thinking-2507 base model. Second, we compare CALM + RL against matched direct distillation baselines: using the same 112-problem SFT set and the same prompt/scaffold, we replace each CALM-corrected target with a complete teacher-generated solution from a much stronger model, and then apply the same downstream RL recipe. Third, we report robustness checks that swap the intervener, test a different model backbone, and evaluate transfer to out-of-domain math benchmarks. Full

run-level provenance for these added experiments is provided in Appendix K.

## 4.2. Main Results

We present the main results in Table 2. First, our method improves the five-benchmark macro-average accuracy from 57.1% to 68.9% (+11.8 points), with especially large gains on challenging benchmarks such as MAMO-Complex (+23.8 points). Second, our compact 4B model reaches a level comparable to 671B DeepSeek-R1-0528 (68.9% vs. 67.5%) and sets a new state-of-the-art on MAMO-Complex (70.3%). These results establish that STORM is a strong final system. The next question is what kind of supervision produces this gain.

## 4.3. Analysis and Ablation Studies

### 4.3.1. WHAT SHOULD THE BASE MODEL LEARN FROM?

We first ask what kind of SFT target is most useful once the base model is already an LRM. Table 3 compares three options on the same five-benchmark evaluation: no SFT, OR-Instruct-3K, and CALM-SFT. The answer is not simply "more worked solutions." OR-Instruct-3K helps the easier benchmarks but collapses on the harder ones, whereas CALM-SFT improves the harder benchmarks while remaining competitive on the easier ones.

### 4.3.2. DOES THE GAIN COME FROM DIRECT DISTILLATION?

We then ask a stronger question: if CALM is replaced by a direct distillation baseline, does the advantage disappear after RL? Table 4 answers this under a matched hard-benchmark control. All three runs use the same base model, the same 112-problem SFT set, the same prompt/scaffold, the same downstream RL recipe, and the same evaluation protocol; only the SFT target differs. For CALM, the target is the base model's own solution after a local repair. For distillation, the target is a complete teacher-generated solution from DeepSeek-R1 or Gemini-2.5-Pro under the same scaffold.

Under this matched setup, CALM + RL achieves a higher average than both distillation baselines. A matched-budget checkpoint view for this controlled run is provided in Appendix L. This is the cleanest evidence for our mechanism claim: the gain is not explained by access to a stronger model alone, but by locally correcting the base model's own solution before RL rather than replacing it with a complete teacher-generated solution.

Building on this calibrated starting point, the RL stage drives a further leap: the five-benchmark macro-average accuracy rises from 58.7% to 68.9% (Figure 5), propelling our 4B model to a level comparable with 671B DeepSeek-R1-0528.

**Table 2.** Main results on optimization modeling benchmarks. **Bold** indicates the best performance in each column. Results marked with * are cited from their original papers. The colored value indicates the absolute performance gain over its base model.

| Models | Model Size | NL4OPT | MAMO Easy | MAMO Complex | IndustryOR | OptMath | Macro AVG |
|---|---|---|---|---|---|---|---|
| *Baseline Models* | | | | | | | |
| GPT-3.5-Turbo | NA | 78.0* | 79.3* | 33.2* | 21.0* | 15.0* | 45.3* |
| GPT-4 | NA | 89.0* | 87.3* | 49.3* | 33.0* | 16.6* | 55.0* |
| DeepSeek-V3 | 671B | 95.9* | 88.3* | 51.1* | 37.0* | 32.6* | 61.0* |
| DeepSeek-R1-0528 | 671B | 86.6 | 78.8 | 69.1 | 52.5 | **50.6** | 67.5 |
| Qwen3-235B-A22B-Thinking-2507 | 235B | 75.8 | 77.2 | 63.6 | **53.2** | 49.6 | 63.9 |
| *Agent-Based Methods* | | | | | | | |
| Chain-of-Experts | NA | 64.2* | - | - | - | - | - |
| OptiMUS | NA | 78.8* | 77.2* | 43.6* | 31.0* | 20.2* | 49.4* |
| *Learning-Based Methods* | | | | | | | |
| LLMOPT-Qwen2.5-14B | 14B | 80.3* | 89.5* | 44.1* | 29.0* | 12.5* | 51.1* |
| ORLM-LLaMA-3-8B | 8B | 85.7* | 82.3* | 37.4* | 38.0* | 2.6* | 49.2* |
| OptMATH-Qwen2.5-7B | 7B | 94.7* | 86.5* | 51.2* | 20.0* | 24.4* | 55.4* |
| SIRL-Qwen2.5-7B | 7B | **96.3*** | **90.0*** | 62.1* | 33.0* | 29.0* | 62.1* |
| *Our Framework: Transforming a 4B LRM* | | | | | | | |
| Qwen3-4B-Thinking-2507 (Base) | 4B | 85.8 | 73.8 | 46.5 | 46.2 | 33.1 | 57.1 |
| STORM-Qwen3-4B (Ours) | 4B | 93.3 +7.5 | 86.3 +12.5 | 70.3 +23.8 | 50.0 +3.8 | 44.5 +11.4 | **68.9** +11.8 |

**Table 3.** Comparison of SFT targets for the same base LRM.

| Benchmark | Base (no SFT) | + OR-Instruct-3K | + CALM-SFT |
|---|---|---|---|
| NL4OPT | 85.8 | **92.9** | 86.6 |
| MAMO Easy | 73.8 | **88.7** | 77.9 |
| MAMO Complex | 46.5 | 40.5 | **54.3** |
| IndustryOR | 46.2 | 27.5 | **44.1** |
| OptMath | 33.1 | 6.6 | **30.4** |
| **Macro AVG** | 57.1 | 51.2 | **58.7** |

**Table 4.** Matched hard-benchmark control for CALM vs. direct distillation.

| Benchmark | CALM + RL | DeepSeek-R1 distill + RL | Gemini-2.5-Pro distill + RL |
|---|---|---|---|
| MAMO Complex | **70.3** | 62.5 | 44.1 |
| IndustryOR | 50.0 | **50.4** | 28.3 |
| OptMath | **44.5** | 29.5 | 20.0 |
| **Hard-3 AVG** | **54.9** | 47.4 | 30.8 |

### 4.3.3. DECONSTRUCTING CALM: AN INSIDE LOOK AT THE CURATION PROCESS

To understand its mechanics, we decompose the CALM data curation process into three phases as summarized in Figure 6: diagnosing common mistakes in the base model's initial solutions, refining them via hinting, and filtering the results into a high-quality SFT dataset.

**Diagnosis of Common Mistakes.** The diagnosis phase identifies failure modes in the base LRM's initial solutions.

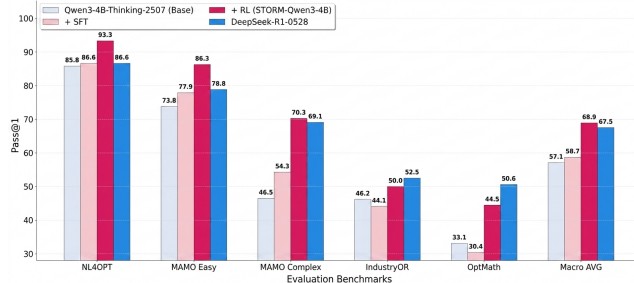

**Figure 5.** Performance evolution across training stages: Base → CALM-SFT → RL (= STORM).

The distribution of interventions (Figure 6, left) reveals two dominant error categories: *Code Utilization Distrust* and *Lack of OR Expertise*. Consistent with our analysis in Section 2.3, the former is more prevalent on the low-to-medium difficulty problems common in our SFT set.

**Refinement via Lightweight Hinting.** The refinement phase uses an iterative hinting loop to correct flawed trajectories. As shown in Figure 6 (middle), this lightweight process, with minimal interventions per problem, significantly boosts the success rate while simultaneously reducing response length. This demonstrates that targeted guidance can enhance both correctness and conciseness.

**Filtering Corrected Solutions.** Finally, the filtering phase ensures only high-quality corrected solutions are used for training. The filtering funnel (Figure 6, right) is highly selective, retaining only solutions that are both correct and deemed clean by the Intervener.

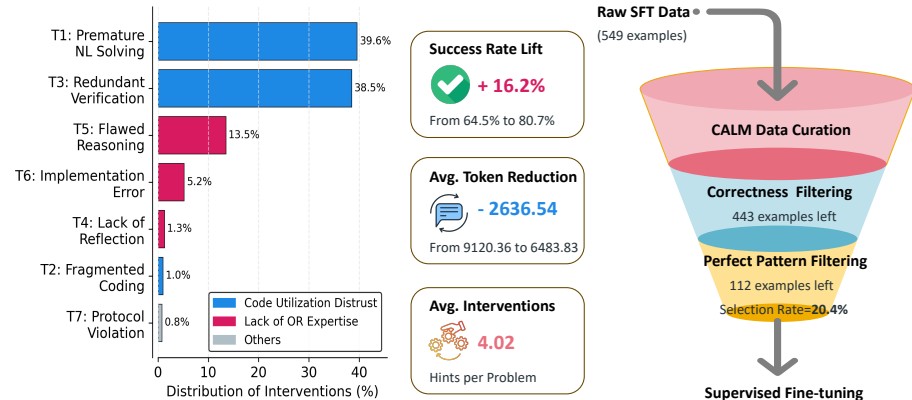

**Figure 6.** The CALM data curation engine.

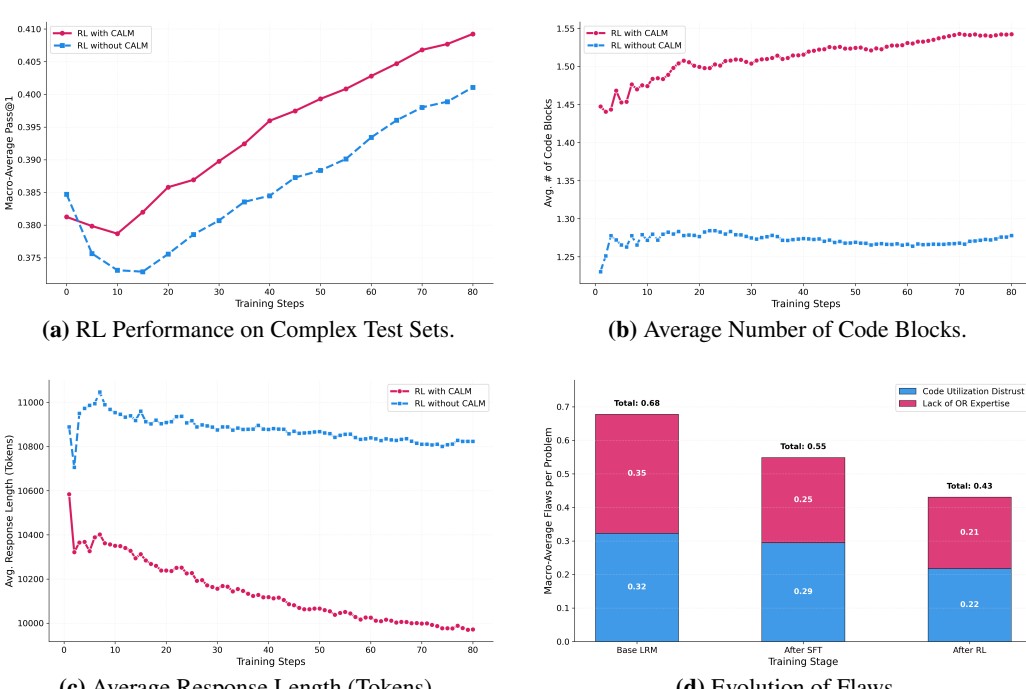

(a) RL Performance on Complex Test Sets.

(b) Average Number of Code Blocks.

(c) Average Response Length (Tokens).

(d) Evolution of Flaws.

**Figure 7.** Behavioral evolution analysis.

### 4.3.4. HOW CALM CHANGES THE RL STARTING POINT

We next study learning dynamics under a controlled comparison between **RL with CALM** and **RL without CALM**. The two runs start from the same base model and use the same RL objective and evaluation protocol; the intended difference is only the SFT target. This experiment is used to study the RL starting point and the observed learning trajectory, not to define a separate final leaderboard. The detailed setup is provided in Appendix E.4.

**Better RL Starting Point.** Figure 7a shows that starting RL from locally corrected solutions has a clear impact. Within the observed training window, RL with CALM stays consistently above the control run and learns more stably. The control model, starting from unguided solutions, learns more

slowly and does not close the gap.

**A Shift Toward Computation-Driven Reasoning.** This is explained by consistent behavioral changes, as shown in Figures 7b and 7c. The RL with CALM model progressively increases its use of code blocks while reducing average response length. This reflects a shift toward expert-like behavior: replacing verbose natural language calculations with concise and reliable code execution. The control model, lacking this guidance, remains verbose and less computation-driven.

**Complementary Roles of SFT and RL.** Figure 7d reveals a clear division of labor. Relative to the base model, *Lack of OR Expertise* decreases from 1.00 to 0.68 after CALM-SFT and to 0.48 after RL, while *Code Utilization Distrust*

decreases from 1.00 to 0.86 after CALM-SFT and to 0.60 after RL. In other words, CALM-SFT mainly corrects modeling and domain-knowledge mistakes, while RL further strengthens the model's habit of relying on solver-based computation. A per-benchmark breakdown is provided in Appendix F.

These gains are accompanied by more reliable solver interaction. From Base to SFT to RL, execution success rises from 91.4% to 92.9% to 96.7%, the share of `Optimal` solver status rises from 42.1% to 55.1% to 62.3%, and the timeout rate falls from 20.2% to 14.3%. Full stage-wise solver statistics are reported in Appendix M.

### 4.3.5. ROBUSTNESS AND SCOPE

We finally test how tightly the effect is tied to one teacher, one backbone, and one benchmark family. As detailed in Appendix N, swapping Gemini-2.5-Pro for open-source DeepSeek-R1-0528 during CALM curation changes the final five-benchmark macro average only from 68.9% to 67.8%, suggesting that the gain is not tied to a single proprietary intervener. We also test CALM on a different model family and scale (Appendix O): DeepSeek-R1-0528-Qwen3-8B improves from 58.4 to 60.2 after CALM-SFT and to 67.8 after CALM-RL. Finally, on three out-of-domain mathematical reasoning benchmarks (Appendix P), RL with CALM remains ahead of RL without CALM (82.33% vs. 80.37% on average). We therefore do not claim universal transfer, but the evidence shows that the effect is not tied only to one proprietary teacher, one 4B base model, or the original OR benchmark suite.

## 5. Conclusion

This paper studies how to adapt a Large Reasoning Model to optimization modeling once the base model already produces long worked solutions of its own. Our central finding is that the training target matters. Fine-tuning on OR-Instruct-3K helps on easier benchmarks but hurts on harder ones. On a five-benchmark comparison of SFT targets, CALM-SFT is stronger than OR-Instruct-3K. On a separate matched hard-benchmark control, replacing CALM with a direct distillation baseline still leads to weaker final RL models. With local correction followed by RL, STORM reaches 68.9% macro-average accuracy across five optimization modeling benchmarks and matches 671B DeepSeek-R1-0528.

Our claim is deliberately narrow. We study optimization modeling with solver-based code execution, not general tool-using agents. Although teacher-swap, cross-model, and out-of-domain math results suggest some transfer, broader tool settings, cheaper curation strategies, and more systematic comparisons to other adaptation regimes remain future work. Because incorrect optimization models can have operational consequences, we view these systems as decision-support tools that require human oversight.

## Acknowledgements

This work is supported in part by the National Natural Science Foundation of China (NSFC) [Grants 72495131, 72394361, 72425013], the Guangdong Provincial Key Laboratory of Mathematical Foundations for Artificial Intelligence (2023B1212010001), the Shenzhen Stability Science Program 2023, the 1+1+1 CUHK-CUHK(SZ)-GDSTC Joint Collaboration Fund (2025A0505000049, 2025A0505000079), the Major Frontier Exploration Program (Grant No. C10120250085) from the Shenzhen Medical Academy of Research and Translation (SMART), the Shenzhen Medical Research Fund (B2503005), and the International Science and Technology Cooperation Center, Ministry of Science and Technology of China (Grant 2024YFE0203000).

## Impact Statement

This work contributes to the advancement of automated optimization modeling, aiming to democratize access to Operations Research tools that have traditionally required specialized expertise. A significant societal benefit of our approach lies in its resource efficiency: by demonstrating that a compact 4B-parameter model can match the performance of models orders of magnitude larger, we highlight a scalable path for deploying expert-level reasoning capabilities in resource-constrained environments.

However, we emphasize that such automated systems should function as decision support tools rather than autonomous decision-makers, particularly in high-stakes domains. As with all large language models, the potential for generating plausible but incorrect logic exists, and human oversight remains essential to ensure safety and reliability. Finally, it is crucial to note that the underlying foundation models may exhibit biases inherent in their pre-training data that were not specifically examined in this study.

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

## A. Related Work

**From Single-Pass to Iterative OR Modeling.** Early learning-based methods, including ORLM (Huang et al., 2025), LLMOPT (Jiang et al., 2024), and SIRL (Chen et al., 2025), trained models to produce a complete solution in a single pass. With modern LRMs (Qwen Team, 2025; DeepSeek-AI, 2025), however, the base model already comes with its own step-by-step way of solving the problem, including writing code, running it, and revising. Our work studies how to adapt these models without replacing the way they already try to solve the task.

**Hint-based Reasoning Adaptation.** Injecting guidance into a model's reasoning is a promising adaptation technique. Concurrent works like START (Li et al., 2025b) use pre-defined, static hint libraries to encourage code use, while CoRT (Li et al., 2025a) relies on manual, human-in-the-loop annotation for its Hint Engineering, limiting scalability. In contrast, our CALM framework introduces a fully automated and dynamic Reasoner–Intervener pattern. It moves beyond static libraries and manual oversight by enabling an expert model to detect flaws and inject tailored hints, providing scalable, process-level correction that keeps most of the base model's own solution intact.

## B. Illustration: Input–Output Structure of Traditional LLMs for Optimization Problems

---

**Example: Optimization Problem in Natural Language and Its Formalization**

**Input (Natural-Language Problem).** A company must transport 25 tons of cargo using trucks, airplanes, or ships. Per-ton costs are \$100, \$120, and \$130, with capacities 10, 20, and 30 tons. Trucks and ships cannot be chosen together. The goal is to minimize the total cost while satisfying the demand.

- - - - - - - - - - - - - - - - - - - - - - - - - - - - - - - - - - - - - - - - - - - - - - - - -

**Output (Mathematical Model and Solver Code).**
*Variables.*

- $x_1, x_2, x_3 \in \{0,1\}$: binary variables indicating whether trucks, airplanes, and ships are selected.

- $y_1, y_2, y_3 \geq 0$: transported volumes (tons) by each mode.

*Objective and Constraints.*

$$\min 100y_1 + 120y_2 + 130y_3$$
$$\text{s.t.} \quad x_1 + x_2 + x_3 \geq 1 \tag{4}$$
$$y_1 \leq 10x_1, \ \ y_2 \leq 20x_2, \ \ y_3 \leq 30x_3 \tag{5}$$
$$x_1 + x_3 \leq 1 \tag{6}$$
$$y_1 + y_2 + y_3 \geq 25 \tag{7}$$

*Program (PuLP).*

```python
from pulp import LpProblem, LpMinimize, LpVariable, LpBinary, lpSum, PULP_CBC_CMD,
    value

# Data
costs = {"trucks":100, "airplanes":120, "ships":130}
caps  = {"trucks":10,  "airplanes":20,  "ships":30}
demand = 25

# Model
m = LpProblem("Transportation", LpMinimize)
x = {k: LpVariable(f"x_{k}", 0, 1, cat=LpBinary) for k in costs}
y = {k: LpVariable(f"y_{k}", 0) for k in costs}

# Objective
m += lpSum(costs[k]*y[k] for k in costs)

# Constraints
m += lpSum(x[k] for k in costs) >= 1
for k in costs:
```

---

```
19      m += y[k] <= caps[k]*x[k]
20 m += x["trucks"] + x["ships"] <= 1
21 m += lpSum(y[k] for k in costs) >= demand
22
23 # Solve
24 m.solve(PULP_CBC_CMD(msg=False))
25 print("Objective:", value(m.objective))
26 for k in costs:
27     print(f"{k}: x={value(x[k])}, y={value(y[k])}")
```

## C. Protocol for Human-in-the-Loop Flaw Taxonomy Creation

This section details the rigorous, multi-stage protocol our team of four human experts (graduate students with OR and STEM backgrounds) followed to establish the seven-flaw taxonomy presented in Section 2.3. The goal was to move from unstructured observations to a systematic and reproducible classification of errors.

**Stage 1: Initial Data Generation and Independent Annotation.** A base LRM (Qwen3-4B-Thinking-2507) was used to generate solutions for a diverse set of 50 problems selected to cover a range of difficulties and types from our benchmark suite. Each of the four annotators independently reviewed these same 50 responses. For each response, they performed an open-ended analysis, identifying and documenting any perceived reasoning errors. Annotators were instructed to assign a descriptive tag (e.g., "manual-calculation-error," "missed-integer-var") and provide a brief textual justification for each identified flaw. This initial stage resulted in four independent sets of annotations, containing a rich but unstructured collection of observed errors.

**Stage 2: Collaborative Clustering and Taxonomy Refinement.** The team then engaged in a collaborative session to synthesize the independent findings. The process was as follows:

1. **Merging**: All unique error tags and justifications from the four annotators were collected into a single master list.

2. **Affinity Clustering**: The team collectively grouped semantically similar tags into higher-level clusters. For example, tags like "manual-calculation-error," "avoids-solver," and "solves-by-hand" were grouped into a cluster that would later become "Premature NL Solving."

3. **Definition and Refinement**: For each cluster, the team collaboratively wrote a precise, operational definition for the flaw type it represented. This process involved several rounds of discussion to ensure the definitions were mutually exclusive and collectively exhaustive for the observed phenomena. Any ambiguous or overlapping clusters were either merged or further refined.

This iterative process led to the convergence on the seven distinct and recurring flaw types detailed in Appendix D. This human-in-the-loop methodology ensures that our taxonomy is grounded in empirical observation and expert consensus.

## D. Trigger Types

As detailed in our protocol (Appendix C), our analysis identified seven recurring flaw types. Six of these are classified as *substantive reasoning flaws* as they represent fundamental errors in the problem-solving process. The seventh, *Protocol Violation*, is classified as a *procedural error* as it relates only to output formatting. Our main analysis in the paper focuses on the six substantive flaws. The definitions for all seven triggers are as follows:

- **[Trigger 1] Premature NL Solving**: After formulating the mathematical model, the LRM starts solving it manually with natural language instead of immediately writing solver code.

- **[Trigger 2] Fragmented Coding**: The LRM writes small, non-executable, or multiple solver-running code blocks instead of a single, comprehensive one.

- **[Trigger 3] Redundant Manual Verification**: After a code output, the LRM manually re-calculates the exact numerical results that were already provided by the solver.

- **[Trigger 4] Lack of Sanity Check/Reflection**: The LRM gets a correct code output but proceeds directly to the final answer without any high-level reflection on the result's plausibility.

- **[Trigger 5] Flawed Reasoning or Modeling**: The LRM's logic is flawed, leading to an incorrect answer. This includes semantic misunderstanding, a wrong mathematical model, or missing constraints (e.g., integers).

- **[Trigger 6] Implementation Error**: The mathematical model is correct, but the code is buggy or does not faithfully represent the model, leading to an incorrect answer.

- **[Trigger 7] Protocol Violation**: The LRM violates a clear instruction, especially regarding the final boxing requirement.

Here, triggers 1-3 exemplify **Code Utilization Distrust**, pinpointing behaviors such as solving problems with natural language instead of code or engaging in inefficient coding practices. Triggers 4-6 are indicators of a **Lack of OR Expertise**, covering fundamental errors in modeling, logical reasoning, and code implementation. A final trigger, Protocol Violation (Trigger 7), serves as a procedural check to ensure the model adheres to specific output formatting instructions. See Table 5 for examples.

**Table 5.** This table illustrates seven common LRM error patterns ('triggers'), showing the original error (red) and analysis of the errors. These triggers include: **(1)Premature NL Solving**, an attempt at manual calculation instead of coding; **(2)Fragmented Coding**, writing separate small code blocks; **(3) Redundant Manual Verification**, unnecessarily re-calculating a solver's result; **(4) Lack of Sanity Check**, failing to reflect on a solution's plausibility; **(5) Flawed Reasoning or Modeling**, formulating an incorrect mathematical model; **(6) Implementation Error**, correctly modeling a question but incorrectly coding a correct model; and **(7)Protocol Violation**, ignoring explicit instructions and embedding the boxed number within a sentence.

---

**Error Type [Trigger 1] Premature NL Solving**: ... I notice that there is only 5 combinations and this is an easy task, so I can just try them all first without writing python code. ... Alternatively, Worker III → B (4), Worker IV → D (3), Worker I → A (9), Worker V → C (7). Total: 4 + 3 + 9 +7 = 23. That's worse. ...

**Analysis**: It incorrectly assumed the problem was a simple task involving only five combinations, thus obviating the need for Python code. The appropriate methodology would be to first construct a mathematical model and then employ a solver, accessed via code, to obtain the solution.

---

**Error Type [Trigger 2] Fragmented Coding**: ... Re-calculating the derivative and solving by hand is unnecessary ... I'll calculate the ratio 'h/r' directly from the output. I can even use another small code block to be precise.

```python
# Calculate the ratio h/r to determine the shape
ratio = h / optimal_r
print(f"Height to radius ratio: {ratio.evalf()}")
```
```output
...
NameError: name 'h' is not defined
```

**Analysis**: This is fragmented coding and will fail because the variables from the first block are not available in the second. The ideal workflow is to perform all related calculations in a single, self-contained solver script.

---

**Error Type [Trigger 3] Redundant Manual Verification**: ... So the code is correct. (But wait, another way: Let's see if there's a possible solution with lower total flavouring. For example, $M = 1, O = 2$ gives 10.) ...

---

**Analysis**: After correctly executing the solver and getting the optimal result, the LRM shows a lack of trust in the tool. It proceeds to manually verify the constraints (which is acceptable) but then attempts to re-solve the problem by hand.

---

**Error Type [Trigger 4] Lack of Sanity Check/Reflection**: ...

```output
Total cost: 24.00
Food 1: 0.0
Food 2: 8.0
...
Food 9: 4.0
```

The output shows that the **optimal solution** uses:

- **8 units of Tofu**

- **4 units of Quinoa**

This combination meets all the nutritional requirements at a **minimum total cost of 24.00**.

**Analysis**: The LRM correctly executes the solver and gets the optimal solution. However, it moves directly from stating the decision variables (8 Tofu, 4 Quinoa) to declaring the final answer without performing a brief sanity check. An expert would quickly verify that this proposed solution indeed satisfies the nutritional constraints and matches the calculated cost.

---

**Error Type [Trigger 5] Flawed Reasoning or Modeling**: ... But maybe in this problem, we can treat them as continuous variables (since the problem might be designed for linear programming without integer constraints) ...

**Analysis**: In the context of this problem, the units of the products must be integers and cannot be continuous. The LRM incorrectly treated continuous variables as integer variables during the modeling process, which led to a flawed model formulation.

---

**Error Type [Trigger 6] Implementation Error**: ...
**Constraints**:
1. **Demand Constraint**: For each store $j$, $\sum_{i=1}^{6} y_{ij} \geq \text{demand}_j$
2. **Capacity Constraint**: For each DC $i$, $\sum_{j=1}^{9} y_{ij} \leq \text{capacity}_i$
3. **Non-negativity**: $y_{ij} \geq 0$ ...

```python
...
# Create flow variables y[i][j] for each DC i and store j
 y = [[pulp.LpVariable(f'y_{i}_{j}', cat='Continuous') for j in range(9)] for i in range(6)]
...
```

```output
 Status: Unbounded
 Total cost: 16417.0
```

...
**Analysis**: Although the LRM was correctly established in the modeling phase with constraints ensuring the non-negativity of all variables $y_{ij}$, this requirement was overlooked during implementation, where the code failed to set a lower bound of zero for $y_{ij}$.

---

**Error Type [Trigger 7] Protocol Violation**: ...
**Final Answer**

> ...I'll now summarize the findings and box the final answer. The optimal solution uses 97.01 square feet of sunflowers and 0 square feet of roses, yielding a maximum profit of $\boxed{43656.72}$.
> **Analysis**: The LRM's final answer formulation violates the instructions. It embeds the boxed number within a sentence, whereas the protocol requires the box to contain only the final numerical answer and be separate from the summary text.

## E. Experimental Details Appendix

### E.1. Benchmark Datasets and Splitting Strategy

Our study utilizes a broad range of public benchmarks (Jiang et al., 2024) for training and evaluation. To ensure a rigorous and unbiased experimental design, we randomly partitioned all available data from eight sources into non-overlapping training (SFT and RL) and test sets. Table 6 provides a comprehensive overview of these sources, their original sizes, and our final partitioning.

While our main evaluation in the paper focuses on five key benchmarks to ensure direct comparability with prior state-of-the-art work (Chen et al., 2025), we provide test splits for all datasets to facilitate future research.

**Table 6.** Comprehensive overview of benchmark datasets and our rigorous splitting into non-overlapping SFT, RL, and Test sets.

| Data Source | | Data Partitioning | | | |
|---|---|---|---|---|---|
| **Benchmark** | **Description** | **Original Size** | **SFT Set** | **RL Set** | **Test Set** |
| NL4Opt | NeurIPS 2022 competition data, focusing on LP formulation. | 46 | 8 | 8 | 30 |
| MAMO-Easy | High-school level MILP problems for fundamental modeling. | 650 | 200 | 350 | 100 |
| MAMO-Complex | Undergraduate-level MILP/LP problems with intricate structures. | 211 | 55 | 56 | 100 |
| IndustryOR | Real-world industrial problems across diverse sectors and types. | 100 | 6 | 12 | 80 |
| OptMath | Challenging mathematical optimization problems for advanced reasoning. | 166 | 30 | 36 | 100 |
| OptiBench | A collection of various optimization problems. | 607 | 250 | 257 | 100 |
| ComplexOR | Complex OR problems from academic and industrial scenarios. | 18 | 0 | 0 | 18 |
| NLP4LP | LP problems sourced from optimization textbooks and lecture notes. | 12 | 0 | 0 | 12 |

### E.2. Implementation Details for the Pilot Study

This section provides the specific implementation details for the pilot study discussed in Section 2.2.

- **Base Large Reasoning Model (LRM):** The LRM used in this study was **Qwen3-4B-Thinking-2507**, a powerful open-source model known for its strong multi-step reasoning capabilities.

- **SFT Dataset:** We used **OR-Instruct-3K** (Huang et al., 2025), a widely-recognized dataset in the field consisting of 3,000 problem-solution pairs written as complete single-pass solutions.

- **Training Procedure:** The base LRM was fine-tuned using a standard supervised fine-tuning (SFT) objective. The training utilized the same set of hyperparameters as our main SFT stage, which are detailed in Table 7.

### E.3. Implementation Details for the CALM & STORM Framework

This section provides a comprehensive overview of the implementation details for our entire framework, including the computing infrastructure, the CALM data curation process, and the two-stage training pipeline.

**Computing Infrastructure.** All experiments were conducted on a cluster of four nodes, each equipped with 8x NVIDIA H800 (80GB) GPUs.

**CALM Data Curation.** The corrected solutions for SFT were generated using our `CALM` framework with the following configuration:

- **Reasoner Model:** `Qwen3-4B-Thinking-2507`.

- **Intervener Model:** `Gemini-2.5-Pro`.

- **Process Control:** The iterative hinting loop was run for a maximum of $N = 5$ interventions per problem. An "intervention" consists of the Intervener identifying a flaw, injecting a hint, and the Reasoner regenerating the trajectory from that point. This limit serves as a practical safeguard to prevent excessively long or unproductive correction cycles. If a trajectory remains flawed after 5 interventions, it is discarded and not considered for the final SFT dataset.

- **Reasoner Generation Parameters:** Temperature set to 0.6, top-p to 0.95. Max response length was 16384 tokens with a maximum of 4 code executions per turn.

- **Intervener Generation Parameters:** Temperature set to 1.0 and top-p to 0.95 to encourage diverse analytical feedback.

**Stage 1: Supervised Fine-Tuning (SFT).** The SFT stage used the 112 corrected solutions curated by the `CALM` process.

- **Base Model:** `Qwen3-4B-Thinking-2507`.

- **Optimizer:** AdamW.

- **Key Hyperparameters:** Summarized in Table 7.

- **Framework:** DeepSpeed Stage 3 with bf16 precision.

**Table 7.** Key hyperparameters for the supervised fine-tuning (SFT) stage.

| Hyperparameter | Value |
|---|---|
| Learning Rate | 1e-5 |
| LR Scheduler | Cosine |
| Warmup Ratio | 0.1 |
| Total Batch Size | 8 |
| Number of Epochs | 3 |
| Max Sequence Length | 22000 |

**Stage 2: Reinforcement Learning (RL).** The RL stage commenced from the final checkpoint of the SFT model, using the following setup:

- **Algorithm:** Group Relative Policy Optimization (GRPO) via the Verl framework (Sheng et al., 2024).

- **Key Hyperparameters:** Detailed in Table 8.

### E.4. Implementation Details for the Controlled Experiment

This section details the setup for the controlled experiment presented in Section 4.3.4, which was designed to isolate the impact of the initial SFT data quality on RL dynamics. The experiment involved a direct comparison between our main model, RL with CALM, and a control model, RL without CALM. To ensure a rigorous comparison, the control model's setup was designed to mirror the main model's in every aspect except for the SFT data.

**SFT Data.** The control model was fine-tuned on the 112 *original, unguided* reasoning trajectories corresponding to the same problems used for the main model's SFT stage.

**Table 8.** Hyperparameters for the Reinforcement Learning stage.

| Hyperparameter | Value |
| --- | --- |
| **General** | |
| Start Model Checkpoint | Final from supervised fine-tuning |
| Learning Rate | 1e-6 |
| $\epsilon$ | 1e-3 |
| Total Epochs | 100 |
| Train Batch Size | 64 |
| PPO Mini-batch Size | 64 |
| KL Loss | Disabled |
| **Rollout Configuration** | |
| Samples per Prompt (N) | 8 |
| Temperature | 0.6 |
| Max Prompt Length | 3000 |
| Max Response Length | 16384 |
| Max Code Execution per Rollout | 4 |

**Hyperparameters.** To maintain a controlled environment, the hyperparameters for the control model's SFT and RL stages were kept identical to those of our main model. Due to computational resource constraints, the RL training for this specific comparative analysis was conducted for 30 epochs. The complete list of hyperparameters for the control model is provided in Appendix E.3 for full transparency.

## F. Detailed Breakdown of Flaw Frequency Evolution

In Section 4.3.4 of the main text, we presented the macro-average trend of flaw frequency reduction. To provide a more granular view, Figure 8 presents a detailed, per-benchmark breakdown of this evolution.

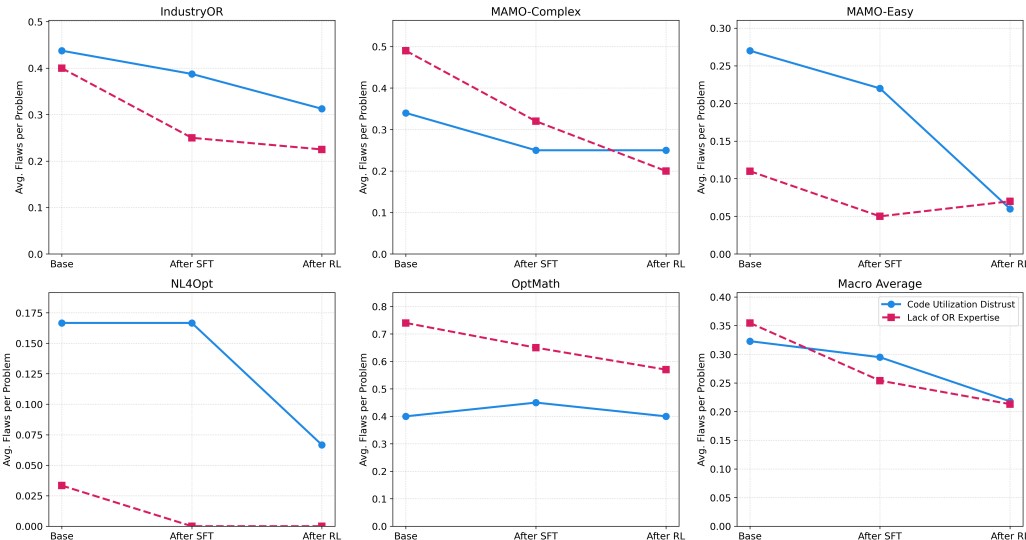

**Figure 8.** A per-benchmark breakdown of the evolution of flaw frequencies. Each subplot shows the average number of flaws per problem for the two main categories across the three training stages: Base LRM, After SFT, and After RL. The 'Macro Average' plot (bottom right) summarizes the general trend.

The six-panel figure illustrates the change in frequency for the two primary flaw categories—*Code Utilization Distrust* (blue solid line) and *Lack of OR Expertise* (red dashed line)—at each training stage.

A detailed analysis of the trends reveals the complementary roles of our two-stage approach:

- **Stage 1 (SFT): Broad-Spectrum Correction.** The supervised fine-tuning stage initiates a significant reduction in both types of flaws across almost all benchmarks. Notably, we observe a substantial drop in the red line (*Lack of OR Expertise*) during this phase (e.g., in `IndustryOR` and `MAMO-Complex`). This suggests that training on locally corrected solutions provides strong initial guidance, helping the model correct fundamental modeling errors and adopt more expert-like problem formulations. The blue line (*Code Utilization Distrust*) also shows a general downward trend, indicating that the model begins to learn more efficient code-use habits.

- **Stage 2 (RL): Targeted Refinement and Mastery.** Building upon the foundation laid by SFT, the reinforcement learning stage continues to refine the model's skills. The RL phase consistently drives down the remaining flaws of both types, pushing the error rates to their lowest levels. This stage allows the model to move beyond simple imitation and achieve a deeper, more robust mastery of both domain knowledge and code use through trial-and-error exploration.

This per-benchmark analysis reinforces our central claim: the two-stage pipeline works synergistically. SFT provides a strong initial correction across the board, and RL builds upon this to achieve a state of expert-level proficiency.

# G. Comparison of Solving Approaches

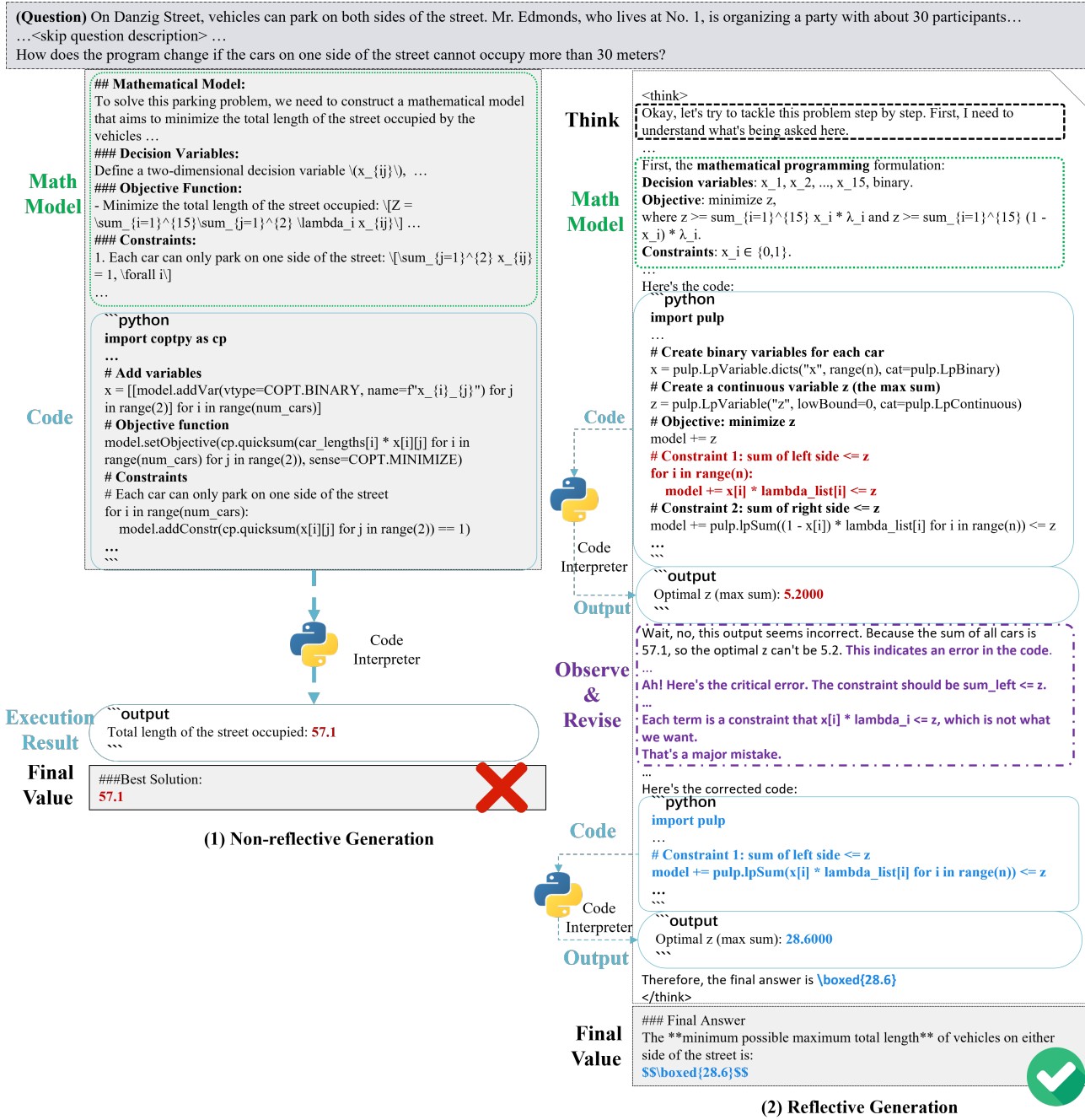

**Figure 9.** An illustrative example comparing single-pass generation (left) and write–run–inspect–revise (right) on a vehicle parking optimization problem.

Figure 9 demonstrates the practical differences between the two solving approaches using a parking optimization task. The single-pass approach (left) formulates a mathematical model and writes the complete code in one step. However, a subtle error in one of the constraints leads to a logically incorrect final answer. Because no execution feedback is used, the model is unable to detect or correct this error.

In contrast, the write–run–inspect–revise approach (right) showcases an iterative process. The model initially generates code that also contains an error, leading to an implausible output. By observing this solver output, the model identifies the flaw in its formulation. It then autonomously corrects the constraint in the code and re-executes it, successfully arriving at the

correct optimal value. This case clearly highlights the advantage of using execution feedback for self-correction.

# H. Intervention process for Specific Flaws

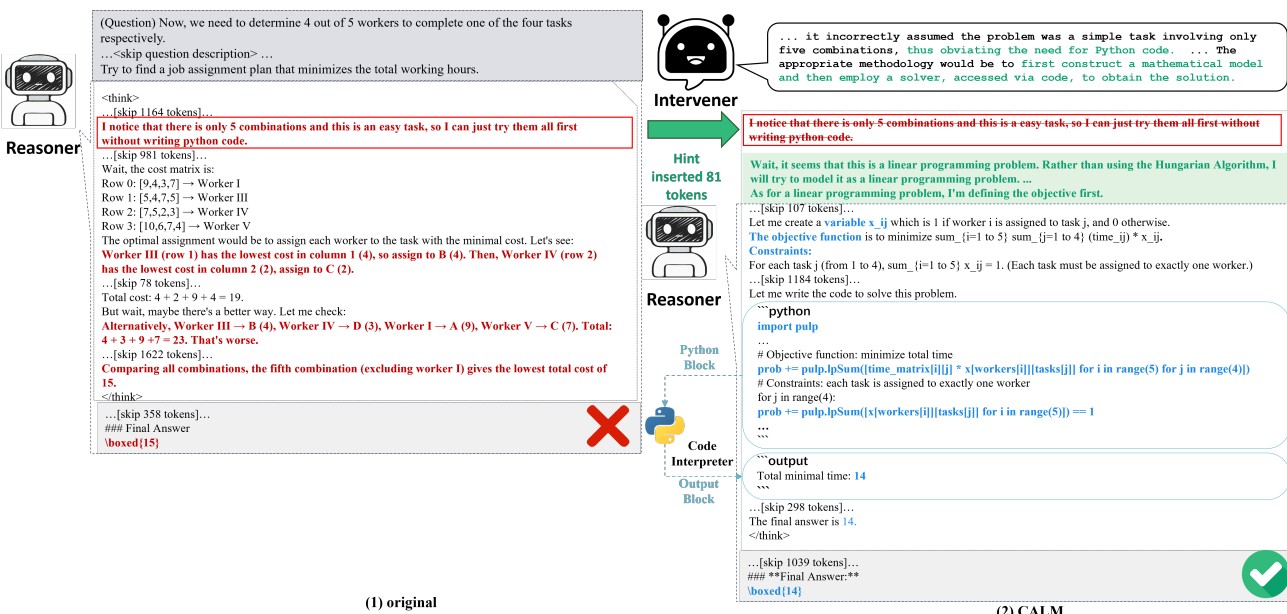

**Figure 10.** A representative *Code Utilization Distrust* error. **Left:** the model exclusively relies on verbal reasoning and fails to use the solver, leading to an incorrect answer. **Right:** after a local hint from **CALM**, the model employs the solver and reaches the correct solution.

# I. Illustrative Case Study of the CALM Framework

To complement our quantitative findings, we provide a case study illustrating how CALM works in practice. Figure 11 traces a multi-turn "Reasoner–Intervener" interaction. The initial trajectory begins with a *Lack of OR Expertise* error, mistaking an Integer Linear Program (ILP) for a standard LP. Even after correction, a secondary issue of *Code Utilization Distrust* emerges. This case demonstrates how a sequence of lightweight, targeted hints can progressively guide the Reasoner through distinct failure modes, ultimately yielding a correct solution without further intervention.

| |
|---|
| **Reasoning** ($\tau^{(0)}$): ... But maybe in this problem, we can treat them as continuous variables (since the problem might be designed for linear programming without integer constraints) ... *Answer:* $\boxed{798.04}$ (Wrong!) |
| **Intervention** ($i = 0$, *Trigger 5 – "Flawed Reasoning or Modeling" detected* ): I've correctly identified the ambiguity around continuous vs. integer variables ... This changes the problem from a Linear Program (LP) to an Integer Linear Program (ILP). ... |
| **Reasoning** ($\tau^{(1)}$): ... So the code is correct. (But wait, another way: Let's see if there's a possible solution with lower total flavouring. For example, $M = 1$, $O = 2$ gives 10.) ... $\boxed{10}$ (Correct!) |
| **Intervention** ($i = 1$, *Trigger 3 – "Redundant Manual Verification" detected* ): Okay, the solver returned M=1, O=2, with a total flavouring of 10. I should trust the solver's optimality result. ... I'll proceed to structure the final answer. |
| **Reasoning** ($\tau^{(2)}$): The solver confirms that the **optimal solution** is: 1 order of **matcha ice cream**; 2 orders of **orange sorbet**... $\boxed{10}$ (Correct!) |
| **Intervention** ($i = 2$, *Trigger 3 – "Redundant Manual Verification" detected* ): ... This result is logical, and I can now confidently format the final answer. |
| **Reasoning** ($\tau^{(3)}$): (correct reasoning without any trigger detected)... $\boxed{10}$ (Correct!) |
| **Intervention** ($i = 3$, *"NO INTERVENTION"* ) |

**Figure 11.** An illustrative case study of the iterative "Reasoner–Intervener" collaboration pattern, where targeted hints progressively correct a flawed reasoning trajectory. Here, red represents the error and blue represents the correction of the Intervener.

### I.1. Prompt Templates

The effectiveness of our framework relies on carefully designed prompts for both the Reasoner's initial task and the Intervener's supervisory role.

**Initial Prompt for the Reasoner.** The Reasoner is initiated with a detailed prompt that outlines the task, reasoning guidelines, tool usage protocols, and the required final answer format. The full template is provided below.

---

**Prompt for Reasoner**

```
Given a mathematical problem, follow the instructions below to solve it.

\#\#\# Instructions:

When solving mathematical problems, you should leverage both natural language
↪   reasoning and Python code execution. Your goal is to provide clear,
↪   detailed explanations while utilizing Python to perform complex
↪   calculations. Follow these guidelines to ensure a coherent and effective
↪   response:

1.  **Natural Language Reasoning:**
    -   Provide comprehensive, step-by-step explanations of your thought
    ↪   process.
    -   Formulate your plan BEFORE writing code. Explain what you are about to
    ↪   do and why.

2.  **Code Execution Rules:**
    -   **Purpose:** Each Python code block must be a complete, self-contained
    ↪   script that executes a single, logical step of your plan.
    -   **Output:** The SOLE mechanism for displaying results is the `print()`
    ↪   function. The purpose of a code block is to compute a value or set of
    ↪   values and explicitly `print()` them for the subsequent `output` block.
```

---

```
      –    **Structure:** Each block must contain all necessary imports and
      ↪    setups. The code must be directly executable. Avoid any boilerplate
      ↪    like `if \_\_name\_\_ == '\_\_main\_\_':`.

3.    **Recommended Toolkit & Best Practices:**
      –    To ensure reliability and environment compatibility, **you must
      ↪    prioritize using the following libraries** for their respective tasks.
      –    For **symbolic mathematics**: use `sympy`.
      –    For **numerical operations**: use `numpy`.
      –    For **scientific computing**: use `scipy`.
      –    For **optimization problems**: use `pulp`.

4.    **Solution Verification and Final Answer:**
      A. **Code Output for Verification:** To ensure your reasoning is
      ↪    transparent and verifiable, your **final code block** should print all
      ↪    key results needed for the solution. For optimization problems, this
      ↪    typically includes:
           *    The optimal objective function value.
           *    The values of the main decision variables.

      C. **Final Answer Formulation:**
           *    **Full Solution Description:** Briefly summarize your findings,
           ↪    referencing the key values printed by your code.
           *    **Final Answer Boxing:** The final step is to put the **single
           ↪    numerical answer** to the main question inside `\\boxed{}`.
                – **Content:** The box should contain **only the number**, without
                ↪    any units, currency signs, or explanatory text.
                – **Example (Correct):** `\\boxed{1234}` or `\\boxed{1234.37}`
                – **Example (Incorrect):** `\\boxed{Total cost is \$1234.0}`

\#\#\# Problem:
{problem_text}
```

**Prompt for the Intervener.** The Intervener is guided by a meta-prompt that defines its role, the ideal expert workflow, and the specific 'Deviation Triggers' it should look for. This prompt is crucial for the automated and targeted nature of our hinting process. The full template is provided below.

### Prompt for Intervener

```
\#\#\# CONTEXT AND GOAL
You are an expert Operations Research (OR) engineer and an LLM Reasoning
↪    Pattern Analyst. Your mission is to assist in generating high-quality
↪    training data for fine-tuning Large Reasoning Models (LRMs).

The ultimate goal is to adapt an LRM's default solution style so that it better
↪    matches the iterative workflow of a human OR expert. The ideal expert
↪    workflow is a cycle of: **1. Understand \& Model -> 2. Code Solver -> 3.
↪    Execute \& Observe -> 4. Reflect \& Debug -> (Repeat)**.

Your specific task is to analyze a given LRM response and, if it deviates from
↪    this ideal workflow, insert a strategic hint to guide it back on track.
↪    This process creates a cleaner corrected solution for training.
```

\#\#\# INSTRUCTIONS
1. First, carefully review the `TASK_DEFINITION` which contains the original
   ↪ problem and instructions given to the LRM.
2. Next, analyze the provided `LLM_RESPONSE_TO_REFINE`.
3. Identify the **first point** of deviation based on the triggers defined
   ↪ below.
4. If a deviation is found, your output MUST be structured using the custom
   ↪ tags `<action>`, `<trigger_type>`, `<analysis>`, `<target_text>`, and
   ↪ `<hint_to_insert>`. The action should be "REPLACE_AND_CONTINUE".
5. The `<target_text>` tag should contain the exact, unique, and contiguous
   ↪ block of text from the original response that needs to be replaced.
6. The `<hint_to_insert>` tag should contain the new hint you've crafted
   ↪ according to the principles below.
7. If the response is ideal, your output should simply be
   ↪ `<action>NO_INTERVENTION</action>`.

### DEVIATION TRIGGERS
* **Trigger 1: Premature NL Solving:** After formulating the mathematical
  ↪ model, the LRM starts solving it manually with natural language instead of
  ↪ immediately writing solver code.
* **Trigger 2: Fragmented Coding:** The LRM writes small, non-executable, or
  ↪ multiple solver-running code blocks instead of a single, comprehensive one.
* **Trigger 3: Redundant Manual Verification:** After a code output, the LRM
  ↪ manually re-calculates the exact numerical results that were already
  ↪ provided by the solver.
* **Trigger 4: Lack of Sanity Check/Reflection:** The LRM gets a correct code
  ↪ output but proceeds directly to the final answer without any high-level
  ↪ reflection on the result's plausibility.
* **Trigger 5: Flawed Reasoning or Modeling:** The LRM's logic is flawed,
  ↪ leading to an incorrect answer. This includes semantic misunderstanding, a
  ↪ wrong mathematical model, or missing constraints (e.g., integers).
* **Trigger 6: Implementation Error:** The mathematical model is correct, but
  ↪ the code is buggy or does not faithfully represent the model, leading to an
  ↪ incorrect answer.
* **Trigger 7: Protocol Violation:** The LRM violates a clear instruction,
  ↪ especially regarding the final boxing requirement.

\#\#\# HINT PRINCIPLES (to guide your hint creation)
* **Be a Guide, Not a Dictator:** Use a first-person, reflective tone (e.g.,
  ↪ "I see, a better way would be...", "Okay, now I should...").
* **Encourage Action:** Frame the hint to prompt a specific, desirable next
  ↪ action.
* **[FOR TRIGGERS 1 \& 2] Force Code Generation:** End your hint with
  ↪ `\n\n\`\`\`python` to strongly encourage immediate and complete code
  ↪ writing.
    * *Example:* "The model is fully formulated. The best next step is to
      ↪ implement this using `pulp` to get an exact solution.\n\n\`\`\`python"
* **[FOR TRIGGER 3] Promote Trust in Tools:** Guide the LRM away from
  ↪ redundant calculation and towards interpretation.
    * *Example:* "The solver has already provided the optimal values.
      ↪ Re-calculating them manually is unnecessary. I should now focus on
      ↪ interpreting the solution."

* **[FOR TRIGGER 4] Encourage Sanity Checks:** Gently guide the LRM to
↪ perform a brief, high-level sanity check. The goal is to cultivate a habit
↪ of reflection, not to force a rigid process.
    * **Hint for Trigger 4 (Lack of Reflection):** "The solver returned an
    ↪ optimal cost of \$392,760. Before I finalize the answer, it's a good
    ↪ practice to quickly reflect on this. Given the high fixed costs of the
    ↪ distribution centers, this value seems to be in a reasonable range.
    ↪ This gives me confidence in the result. Now, I'll proceed to format the
    ↪ final solution."
    * **[Alternate Hint with Code-Assisted Check]:** "The solver returned an
    ↪ optimal cost of \$392,760. That seems plausible. To build more
    ↪ confidence, I could write a quick script to explore a simplified
    ↪ scenario, like checking the cost if I only open the three cheapest
    ↪ centers. This will help verify my understanding.\n\n\`\`\`python"
* **[FOR TRIGGERS 5-7] Inject Focused Expertise:** Craft a concise hint that
↪ addresses the specific flaw found.
    * *Hint for Trigger 5 (Model Completeness Error):* "I've noticed the
    ↪ solution provides a fractional number of cars, which isn't practical.
    ↪ This suggests I missed an integer constraint in my original model. I
    ↪ should correct this by redefining the variables as integers in my code
    ↪ and re-running it."
    * *Hint for Trigger 6 (Implementation Error):* "I've spotted a bug. My
    ↪ math model for the constraint was `A <= B`, but in the code I wrote `A
    ↪ >= B`. I need to correct this implementation error to match my model."

\#\#\# OUTPUT STRUCTURE (MUST use these custom tags)
<action>REPLACE_AND_CONTINUE</action>
<trigger_type>[Trigger 1 | Trigger 2 | ... | Trigger 7]</trigger_type>
<analysis>[A brief explanation of why this intervention is necessary based on
↪ the detected trigger]</analysis>
<target_text>[The exact text from the original response to be
↪ replaced]</target_text>
<hint_to_insert>[Your newly crafted hint goes here]</hint_to_insert>

(OR, if no intervention is needed)

<action>NO_INTERVENTION</action>

\#\#\# --- START OF TASK ---

\#\#\# TASK_DEFINITION:
```text
{task_definition}
```

\#\#\# GROUND_TRUTH_ANSWER (if available)
The known correct final answer for the objective function is:
↪ `\\boxed{[ground_truth_answer]}`

You should use this ground truth to definitively verify the numerical
↪ correctness of the LRM's final boxed answer. If the LRM's answer is
↪ incorrect, your primary goal is to identify the root cause of the
↪ discrepancy.

```text
\#\#\# LLM RESPONSE TO REFINE:
```text
{llm_response_text}
```

## J. Flaw Quantification of the Base LRM

To achieve a scalable and consistent analysis across thousands of model responses, we utilized Gemini-2.5-Pro as an expert annotator. Its task was to classify flaws in the base LRM's generated solutions based on the seven pre-defined categories described in Appendix D.

**Distinction from the CALM Intervener.** It is crucial to distinguish this analytical use of an external model from its role as the dynamic **Intervener** within our CALM data generation framework (Section 3.2).

- **For Quantification (here):** The model acts as a **static classifier**. Its goal is to analyze a completed response and output a structured list of detected flaws for measurement purposes. It does not interact with the LRM.

- **For CALM Intervention (Section 3):** The model acts as an **interactive agent**. Its goal is to monitor a reasoning process in real-time and inject corrective hints to guide the LRM towards a better solution, thereby generating new training data.

While both roles leverage the same underlying understanding of OR modeling flaws, their functions and objectives within our study are entirely separate.

**Prompt for Flaw Classification.** The prompt below was used to guide the Gemini-2.5-Pro model in its role as a static classifier.

---

**Prompt for Flaw Classification**

```
### CONTEXT AND GOAL
You are an expert Operations Research (OR) engineer and an LLM Reasoning
↪  Pattern Analyst. Your mission is to assist in generating high-quality
↪  training data for fine-tuning Large Reasoning Models (LRMs).

The ultimate goal is to analyze an LRM's solution and determine whether it
↪  departs from the iterative workflow of a human OR expert. The ideal expert
↪  workflow is a cycle of: **1. Understand & Model -> 2. Code Solver -> 3.
↪  Execute & Observe -> 4. Reflect & Debug -> (Repeat)**.

Your specific task is to analyze a given LRM response and identify which
↪  predefined triggers appear in it.

### INSTRUCTIONS
1.  First, carefully review the `TASK_DEFINITION`, which contains the original
↪  problem and instructions given to the LRM.
2.  Next, analyze the provided `LLM_RESPONSE_TO_REFINE`.
3.  Identify at most two deviations based on the triggers defined below.
4.  If a deviation is found, your output MUST use the custom tag
↪  `<trigger_type>`.
5.  The detected trigger(s) should appear inside `<trigger_type>`. For example,
↪  `<trigger_type>Trigger 1</trigger_type>`. If there are multiple triggers,
↪  separate them with `;`, for example, `<trigger_type>Trigger 1;Trigger
↪  7</trigger_type>`.
```

```
6.  If the response is ideal, your output should simply be
↪   `<trigger_type>Correct</trigger_type>`.

### DEVIATION TRIGGERS
*   **Trigger 1: Premature NL Solving:** After formulating the mathematical
↪   model, the LRM starts solving it manually with natural language instead of
↪   immediately writing solver code.
*   **Trigger 2: Fragmented Coding:** The LRM writes small, non-executable, or
↪   multiple solver-running code blocks instead of a single, comprehensive one.
*   **Trigger 3: Redundant Manual Verification:** After a code output, the LRM
↪   manually re-calculates the exact numerical results that were already
↪   provided by the solver.
*   **Trigger 4: Lack of Sanity Check/Reflection:** The LRM gets a correct code
↪   output but proceeds directly to the final answer without any high-level
↪   reflection on the result's plausibility.
*   **Trigger 5: Flawed Reasoning or Modeling:** The LRM's logic is flawed,
↪   leading to an incorrect answer. This includes semantic misunderstanding, a
↪   wrong mathematical model, or missing constraints (e.g., integers).
*   **Trigger 6: Implementation Error:** The mathematical model is correct, but
↪   the code is buggy or does not faithfully represent the model, leading to an
↪   incorrect answer.
*   **Trigger 7: Protocol Violation:** The LRM violates a clear instruction,
↪   especially regarding the final boxing requirement.

### OUTPUT STRUCTURE (MUST use these custom tags)
<trigger_type>[Trigger 1 | Trigger 2 | ... | Trigger 7];...;[Trigger 1 |
↪   Trigger 2 | ... | Trigger 7]</trigger_type>

### --- START OF TASK ---

### TASK_DEFINITION:
```text
{task_definition}
```
### LLM RESPONSE TO REFINE:
```text
{llm_response_text}
```
```

**Validation of the LLM Annotator.** To ensure the reliability of the automated quantification process, we validated the LLM annotator's performance against human labels. We randomly sampled 30 responses from the test set, which were independently annotated by both the LLM (using the aforementioned prompt) and one of our expert human annotators.

The agreement between the LLM and human labels was then measured. The LLM achieved an accuracy of 93.3% in identifying and correctly classifying the flaw types present in the responses, calculated based on the instance-level matching of flaw categories. This high level of agreement provides strong evidence for the validity of using the LLM for scalable and consistent flaw quantification across the entire benchmark suite.

## K. Run-Level Provenance for Matched Distillation Controls

This section provides the run-level provenance for the matched direct-distillation controls reported in Table 4. In the core CALM-vs-distillation comparison, the only intended change is the SFT target. Everything else is held fixed so that the added controls isolate the mechanism of interest: localized repair of the base model's own solution vs. full teacher-solution replacement.

**Table 9.** What is held fixed across matched control runs.

| Shared Setting | Value |
|---|---|
| Base model / initialization | Qwen3-4B-Thinking-2507, same initialization |
| SFT set | Same 112-problem CALM SFT set |
| SFT train size | 112 trajectories for each run |
| Prompt template / scaffold | Same prompt template, max 4 code executions per turn |
| SFT hyperparameters | Same as Table 7 |
| RL initialization rule | RL starts from the final SFT checkpoint of each run |
| RL training data | Same RL-pool partition (Table 6) |
| RL objective / reward | Same GRPO training, same binary correctness reward (Eq. 3) |
| RL hyperparameters | Same as Table 8 |
| Evaluation | Same 8-sample pass@1 (temp=0.6, top-p=0.95), max 4 executions |
| Hardware | Same training environment |

For the two distillation baselines (DeepSeek-R1 distill + RL and Gemini-2.5-Pro distill + RL), we keep the same 112-problem SFT set and replace only the SFT target with a complete teacher-generated solution from the corresponding stronger model under the same prompt/scaffold.

## L. Matched-Budget RL Checkpoint View

To complement the final-performance comparison in Table 4, we track the aggregate checkpoint score at intermediate RL training steps for all three runs. This provides a view of the learning dynamics under the same RL budget.

**Table 10.** Matched-budget checkpoint scores during RL. All runs use the same base model, the same RL recipe, and the same controlled evaluation slice; only the SFT target differs. These values are intended to show learning dynamics and are not directly comparable to the final Hard-3 results in Table 4.

| RL Step | CALM + RL | DeepSeek-R1 distill + RL | Gemini-2.5-Pro distill + RL |
|---|---|---|---|
| 0 | 0.455 | 0.464 | 0.273 |
| 20 | 0.477 | 0.464 | 0.293 |
| 40 | 0.493 | 0.466 | 0.299 |
| 60 | 0.489 | 0.468 | 0.318 |
| 80 | 0.503 | 0.472 | 0.319 |

After a brief initial gap at Step 0, CALM + RL overtakes both distillation baselines from Step 20 onward and stays above them within the observed training window. These checkpoint values are reported only to show learning dynamics in this controlled run.

## M. Stage-Wise Solver Interaction Statistics

Table 11 reports code executability and solver-status distributions across training stages, evaluated on the five main benchmarks.

**Table 11.** Code executability and solver-status distribution across training stages.

| Stage | Code Use | Exec Success | Optimal Status | Timeout |
|---|---|---|---|---|
| Base | 92.2% | 91.4% | 42.1% | 20.2% |
| After CALM-SFT | 89.4% | 92.9% | 55.1% | 22.2% |
| After RL (= STORM) | 95.5% | 96.7% | 62.3% | 14.3% |

RL both increases code utilization and makes solver interaction more reliable, with the Optimal rate rising from 42.1% to 62.3% while Timeout drops from 20.2% to 14.3%.

## N. Intervener Swap Robustness

As a robustness check on intervener dependence, we swapped only the intervener during CALM curation while keeping the raw 549-problem curation pool, the intervention budget ($N = 5$), the CALM filtering rule, and all downstream SFT/RL/evaluation settings fixed.

**Table 12.** Intervener swap robustness. Changing the intervener has minimal impact on the final five-benchmark macro average.

| Intervener | Edited-Token Ratio | Retained Solutions | Final Macro Avg |
|---|---|---|---|
| Gemini-2.5-Pro | $\sim$2.6% | 112 | 68.9 |
| DeepSeek-R1-0528 | $\sim$2.8% | 55 | 67.8 |

Despite Gemini achieving a higher correction rate during curation (49.7% vs. 29.0% on initially wrong cases), the final downstream performance is nearly identical, confirming that CALM is robust to the choice of intervener.

## O. Cross-Model Generalization

We tested the CALM pipeline on a different model family and scale to check whether the two-stage pattern generalizes beyond Qwen3-4B.

**Table 13.** Cross-model generalization. Five-benchmark macro average on a different model backbone.

| Model | Base | + CALM-SFT | + CALM-RL |
|---|---|---|---|
| DeepSeek-R1-0528-Qwen3-8B | 58.4 | 60.2 | 67.8 |

On this different backbone, CALM again provides a useful SFT initialization and a substantially stronger final RL model.

## P. Out-of-Domain Mathematical Reasoning

To test whether the benefit extends beyond the OR benchmarks used in training, we evaluated the final CALM-trained model (RL with CALM) and the matched control model (RL without CALM) on three out-of-domain mathematical reasoning benchmarks.

**Table 14.** Out-of-domain math evaluation. RL with CALM vs. RL without CALM on benchmarks never used in training.

| OOD Benchmark | RL without CALM | RL with CALM |
|---|---|---|
| AIME 2025 | 68.33 | 69.58 |
| AMC 2023 | 91.71 | 94.12 |
| MATH500-Level5 | 81.06 | 83.30 |
| Average | 80.37 | 82.33 |

The OOD gain is also behavioral: on the same benchmarks, the CALM-trained model uses executable code more actively (1.06 vs. 0.73 code blocks per problem on average), consistent with the mechanism emphasized in the paper. Notably, while the CALM SFT data is dominated by the optimization-specific library `pulp` (93.3% of traces), the OOD math traces are dominated by `sympy` (70.1%), with `pulp` nearly absent. This suggests that CALM encourages task-adaptive tool selection rather than overfitting to one fixed workflow.

