# OpenReview forum: "CALM Before the STORM: Unlocking Native Reasoning for Optimization Modeling"
_ICML.cc/2026/Conference — ICML 2026 regular_

### Official Review · Reviewer_5hWc · 2026-03-05

**Soundness:** 3
**Presentation:** 3
**Significance:** 3
**Originality:** 3
**Overall Recommendation:** 5
**Confidence:** 3

**Summary:**

This paper studies how to adapt large reasoning models to optimization modeling tasks. The authors first show that directly finetuning LRMs on traditional non-reflective datasets actually hurts performance on complex tasks, because it overwrites the model's native reasoning patterns with a rigid generation style. To address this, they propose CALM, a framework where an expert intervener identifies reasoning flaws in the LRM's trajectories and injects lightweight corrective hints to produce improved trajectories. These corrected trajectories are filtered for quality and used in a two-stage training pipeline: first SFT for soft adaptation, then RL with GRPO for autonomous mastery. The final model STORM is a 4B parameter model that achieves 68.9% macro average accuracy across five optimization modeling benchmarks, matching the performance of 671B DeepSeek-R1-0528. The paper also provides a taxonomy of 7 reasoning flaws grouped into two categories: code utilization distrust and lack of OR expertise.

**Compliance With Llm Reviewing Policy:**

Affirmed.

**Key Questions For Authors:**

See weaknesses

**Limitations:**

Yes

**Strengths And Weaknesses:**

Strengths:
(1) Significance: the core observation is very valuable. Finetuning LRMs to improve performance is a practical problem.
(2) Originality: the framework proposes a clever approach to data synthesis by correcting the model's own flawed reasoning trajectories with minimal interventions.
(3) Presentation:  the paper is easy to follow

Weakness:
(1) the paper focuses on optimization modeling which is a relatively niche domain. I‘d like to see more experiments on generalization of this method

---

> ### Author Rebuttal · Authors · 2026-03-29
>
> We thank the reviewer for the positive feedback and the constructive suggestion on generalization. We appreciate the recognition of the practical importance of adapting LRMs to optimization modeling, the originality of lightweight corrections on model-native trajectories, and the clarity of the presentation. The key remaining question concerns scope, namely whether CALM extends beyond optimization modeling. To address this, we conduct additional targeted analyses.
>
> Our conclusion is positive but carefully scoped. CALM is effective not only on OR benchmarks, but also transfers to a new domain, a different student backbone and scale, and tool-use behaviors that are consistent with the proposed mechanism.
>
> ### 1. Out-of-Domain Generalization Beyond Optimization Modeling
>
> To directly test whether the benefit is confined to OR, we evaluated the final CALM-trained model (`RL with CALM`, STORM) and the matched control model (`RL without CALM`) on three OOD mathematical reasoning benchmarks that were never used in OR training.
>
> | OOD Benchmark | RL without CALM | RL with CALM | Lift |
> |---|---:|---:|---:|
> | aime25 | 68.33 | **69.58** | +1.25 |
> | amc23 | 91.71 | **94.12** | +2.41 |
> | math500-level5 | 81.06 | **83.30** | +2.24 |
> | Average | 80.37 | **82.33** | **+1.96** |
>
> This directly addresses the reviewer's question: the benefit is not confined to the OR benchmarks used in training; it transfers consistently to a different reasoning domain.
>
> ### 2. The Transferred Gain Is Behavioral, Not Just Numerical
>
> We further analyzed how the OOD gain is achieved. On the same OOD math benchmarks, the CALM-trained model uses executable code more actively:
>
> | Behavioral Metric on OOD Math | RL without CALM | RL with CALM | Change |
> |---|---:|---:|---:|
> | Avg. code blocks per problem | 0.73 | **1.06** | **+46%** |
>
> This matches the mechanism emphasized in the paper. CALM is designed to reduce `Code Utilization Distrust`, i.e., the tendency to avoid reliable external computation even when it would help. The OOD behavior suggests that this correction is not specific to OR alone.
>
> ### 3. Evidence for Task-Adaptive Tool Selection
>
> We also compared the tool/library usage pattern between CALM SFT data and the OOD math evaluation traces.
>
> In the CALM SFT data, library usage is dominated by the optimization-specific library `pulp`:
>
> | Library usage in CALM SFT data | % of traces containing this library |
> |---|---:|
> | `pulp` | **93.3%** |
> | `sympy` | 10.5% |
> | others | small |
>
> (Percentages reflect per-trace presence and can overlap when a trace imports multiple libraries.)
>
> By contrast, on the OOD math benchmarks, the dominant library becomes `sympy`, while `pulp` is nearly absent:
>
> | Library usage on OOD math traces | % of traces containing this library |
> |---|---:|
> | `sympy` | **70.1%** |
> | `pulp` | negligible |
>
> We interpret this conservatively. The point is not that CALM teaches a previously unseen tool from scratch. Rather, the model's tool preference shifts with the task domain: the training traces are optimization-heavy, whereas the OOD math traces are dominated by symbolic-math usage. This is consistent with CALM encouraging task-adaptive tool selection rather than overfitting the model to one fixed optimization workflow.
>
> ### 4. Generalization Across Student Backbones / Scales
>
> We also tested CALM on a different student family / scale:
>
> | Student | Base | + CALM-SFT | + CALM-RL |
> |---|---:|---:|---:|
> | DeepSeek-R1-0528-Qwen3-8B | 58.4 | 60.2 | **67.8** |
>
> This suggests that the two-stage CALM pattern is not tied to one specific student. On a different backbone, CALM again provides a useful SFT initialization and a substantially stronger final RL model.
>
> ### 5. What This Supports, and What It Does Not Yet Claim
>
> Taken together, these results support a focused generalization claim:
>
> - across domains: optimization -> OOD math,
> - across behaviors: stronger reliance on external computation in new tasks,
> - across students: similar two-stage gains on a different backbone/scale.
>
> We therefore do not claim generalization to all external-tool settings or broad multi-tool agents. The narrower conclusion supported here is that CALM teaches a transferable form of tool-augmented self-correction that extends beyond the original OR benchmark family.
>
> In the revision, we will add the OOD math results, the OOD code-use behavior analysis, the task-adaptive library-usage analysis, and the cross-student-family result. We are grateful to the reviewer for raising this point, which helped us strengthen the paper into a more general statement about transferable tool-augmented reasoning.

---

> > ### Author Rebuttal · Reviewer_5hWc · 2026-03-31
> >
> > The author has resolved my questions in the rebuttal

---

> > > ### Author Response · Authors · 2026-04-01
> > >
> > > Thank you again for the thoughtful review and for confirming that the added analyses **fully addressed** your concern.
> > >
> > > We especially appreciate your early recognition of the practical significance of adapting LRMs to optimization modeling, the originality of lightweight correction on student-native trajectories, and the clarity of the presentation. Your suggestion to strengthen the scope/generalization discussion directly helped us improve the paper.
> > >
> > > In response, we added the out-of-domain math evaluation, the OOD code-use behavior analysis, the task-adaptive library-usage comparison, and the cross-student-family result. These additions helped us broaden the evidence beyond optimization modeling and sharpen the scope of the paper into a more careful claim about **transferable tool-augmented reasoning**.
> > >
> > > We are genuinely grateful for this suggestion, which **materially improved the final paper**. We will incorporate these analyses and the corresponding scope clarification in the revision.

---

### Official Review · Reviewer_gpH7 · 2026-03-11

**Soundness:** 1
**Presentation:** 2
**Significance:** 1
**Originality:** 1
**Overall Recommendation:** 3
**Confidence:** 3

**Summary:**

This paper proposes a new approach to synthesizing SFT datasets for optimization modeling tasks. Previously, people generated non-reflective data to train the model. However, this paper states that it does not provide LLMs with the opportunity to receive feedback and self-refine, and that it also does not align with the base model's native/original reasoning pattern for fine-tuning. Therefore, this paper uses a reasoner-intervener model, where the reasoner generates training trajectories on its own, and an intervener identifies flaws and hints to the reasoner to generate correct ones. As a result, the 4B model tuned by the datasets generated by this new method achieves better performance than baselines, and also achieves better performance when applying RL afterwards.

**Compliance With Llm Reviewing Policy:**

Affirmed.

**Final Justification:**

- The additional experimental results provided in the paper mostly address the concerns about fairness of the comparisons, including the one comparing with direct teacher-student comparison. Though these results are very important, so they are expected to be presented in the original paper.
- The concern about novelty is partially addressed, if this methodology is indeed new for solving OR tasks. Though, the technique is not very novel in the general ML domain.
Given the additional results provided, the assessment is updated from 2 to 3.

**Key Questions For Authors:**

N/A

**Limitations:**

See the weaknesses stated above.

**Strengths And Weaknesses:**

Soundness:
- Advantages:
  - The reasoner-intervener model is reasonable and plausible to generate SFT data.
  - The experimental results show that the generated datasets can indeed improve performance with SFT and RL.
  - Multi-time sampling is used in the evaluation to ensure reliable results.

- Weaknesses:
  - The narrative of "Non-reflective Generation" vs "Reflective Generation" is weird.
    - Reflective Generation seems to be the case where LLMs can do trial-and-error. Such a self-refinement loop was proposed and well-studied by previous works, and cannot be a novelty of this paper.
    - The paper seems to mix up self-refinement with extended thinking/reasoning/test-time scaling. Reasoning models typically refer to models like o1 and its following similar works like R1, which have an extended CoT reasoning before giving the final answer, and the reasoning may have self-reflections, but here they don't necessarily receive feedback from environments by tool calling. Receiving feedback from environments and reflecting on it is not a unique capability of reasoning models. "Non-thinking models" can also do this procedure. The paper cites Qwen3 and DeepSeek-R1 for "Large Reasoning Models", mixing up reflection on environment-based feedback and thinking/reasoning performed by models themselves unnecessarily with environments, leading to a defective story.
    - Maybe the real novelty of this paper is the "localized/targeted" modification/refinement to the base-model-generated training trajectories. But generating training trajectories with privileged information is also an already explored technique.

  - There is no comparison to the most straightforward teacher-student distillation approach, i.e., just training the base model on the trajectories generated from the stronger teacher model (Gemini-2.5-Pro) with rejection sampling. Without such results, it cannot justify why we need to train on refined student trajectories, instead of the ones directly generated by stronger models, as both rely on a stronger teacher model.
  - The experimental setup is outdated. Modern flexible agentic scaffolds are not used for evaluation, such as Claude Code/OpenCode. Limiting the code execution time to 4 is unrealistic, as those agents can usually issue many tool calls. Limiting the context length would be more reasonable.
  - The comparison with models trained with datasets synthesized by other models is meaningless. If some datasets are synthesized with weaker models than Gemini-2.5-Pro, then it is quite predictable that they are worse.
  - In Figure 7a, it is hard to say that the red curve is steeper than the blue one, and it is hard to predict the final converged performance of both curves (e.g., they may converge to the same level of performance).


Presentation:
- Advantages:
  - There are case studies and a taxonomy helping to understand how a difference is made.

- Weaknesses:
  - Same as above, the way of presenting/framing the approach is not sound due to the mixing up of reasoning and environment-based reflection.


Significance and Originality:
- Generally, training a model with trajectories that maintain its original patterns is an interesting topic to explore.
- However, this paper does not touch on the core challenges and discussions around this direction (on-policy data vs off-policy data vs nearly-on-policy data), such as what kind of modification to the student trajectories yields the best performance, how to make minimal modifications if needed, how to alleviate the reliance on stronger teacher models to scale up this method, etc.
- Due to the defective narrative and experimental design, the lessons learned from this paper on how to improve model performance by data synthesis can be quite limited.

---

> ### Author Rebuttal · Authors · 2026-03-28
>
> We thank the reviewer for the careful reading; we revise our position below. We use **W1** for **Weakness 1**, etc.
>
> **Starting point.** We respectfully clarify that our work is not a straightforward transfer or superficial integration. Instead, we study LLMs for optimization modeling with broad applications. Our paper identifies a key phenomenon: many methods for enhancing LLMs fail in this domain. In our pilot study (Section 2), we identify underlying structural issues, which serves as the foundation of our work. Based on this, we design CALM, which intervenes only when execution reveals concrete errors while preserving the remaining reasoning process.
>
> > **W1.** *"The narrative of 'Non-reflective Generation' vs 'Reflective Generation' is weird..."*
>
> We accept this criticism and will replace the terminology with **single-pass generation** vs. **execution-informed iterative reasoning**.
>
> We do **not** claim reflection is novel. Our claim is narrower: adaptation is stronger when student trajectories are **locally corrected rather than replaced**.
>
> 1. **Not arbitrary.** Common alternatives (full-trace replacement, reflective SFT, distillation) all fail on complex OR tasks.
>
> 2. **Mechanistically different.** Prior methods replace the student's trajectory entirely. CALM injects corrections at ~2.6% of tokens, **preserving >97.4%**, and the student resumes its own reasoning.
>
> 3. **Empirical evidence.** OptiMUS — a representative agent-based self-correction method — on the same backbone:
>
> | Method | Macro Avg |
> |---|---:|
> | Qwen3-4B-Thinking-2507 (base) | 57.1 |
> | OptiMUS (Qwen3-4B-Thinking-2507) | 60.4 |
> | STORM | **68.9** |
>
> OptiMUS improves only marginally (+3.3), while STORM achieves +11.8, confirming that generic self-correction without execution-grounded localized intervention is insufficient for OR.
>
> > **W2.** *"No comparison to teacher-student distillation..."*
>
> We ran a matched comparison (same student, same RL stage):
>
> | Method | MAMO-Complex | IndustryOR | OptMath | Avg |
> |---|---:|---:|---:|---:|
> | CALM + RL (=STORM) | 70.3 | 50.0 | 44.5 | 54.9 |
> | DeepSeek-R1 distill + RL | 62.5 | 50.4 | 29.5 | 47.4 |
> | Gemini distill + RL | 44.1 | 28.3 | 20.0 | 30.8 |
>
> CALM + RL outperforms both distillation baselines. Full-trace distillation disrupts the student's native reasoning style, degrading solver-interaction patterns. Moreover, **compact open-source LLMs are critical for real OR deployment** due to privacy, cost, and on-premise constraints.
>
> > **W3.** *"Experimental setup is outdated... 4 execution limit unrealistic..."*
>
> Our evaluation targets **OR optimization modeling**, not general coding agents. The 4-execution limit is a standardized budget — on our final model, **98.5%** of rollouts finish within 4 executions and **97.3%** of successful ones before the 4th. A fair comparison also requires a **fixed scaffold**; mixing agentic frameworks would conflate scaffold quality with data quality. We will clarify this in the revision.
>
> > **W4.** *"Comparison with models trained with other synthesized datasets is meaningless..."*
>
> We replaced Gemini-2.5-Pro with open-source DeepSeek-R1-0528, keeping the pipeline fixed:
>
> | Intervener | Intervener ratio | Final Performance |
> |---|---:|---:|
> | Gemini-2.5-Pro | ~2.6% | 68.9 |
> | DeepSeek-R1-0528 | ~2.8% | 67.8 |
>
> The two teachers behave very differently during intervention:
>
> | Teacher-side hinting statistic | Gemini-2.5-Pro | DeepSeek-R1-0528 |
> |---|---:|---:|
> | Initial problem accuracy | 64.48% | 66.12% |
> | Final accuracy after iterative hinting | 80.69% | 75.23% |
> | Correction rate for initially wrong cases | 49.74% | 29.03% |
> | Avg. hints / problem | 4.02 | 4.37 |
>
> Despite Gemini achieving a much higher correction rate (49.7% vs. 29.0%), the final student performance is nearly identical (68.9 vs. 67.8), confirming that CALM is robust to teacher quality.
>
> > **W5.** *"Figure 7a: hard to say the red curve is steeper..."*
>
> We provide a matched-budget checkpoint view:
>
> | Step | CALM+RL | DeepSeek-R1 distill+RL | Gemini distill+RL |
> |---:|---:|---:|---:|
> | 0 | 0.455 | 0.464 | 0.273 |
> | 20 | 0.477 | 0.464 | 0.293 |
> | 40 | 0.493 | 0.466 | 0.299 |
> | 60 | 0.489 | 0.468 | 0.318 |
> | 80 | 0.503 | 0.472 | 0.319 |
>
> We agree the asymptotic behavior cannot be determined from Figure 7a alone and will soften the claim. The primary evidence is the **final-performance gap**: at Step 80, CALM+RL reaches 0.503 vs. R1 distill+RL 0.472 and Gemini distill+RL 0.319. On the three hardest benchmarks, CALM+RL (54.9) outperforms R1 distill+RL (47.4) and Gemini distill+RL (30.8) by large margins.
>
> > **W6.** *"No discussion of on-policy vs off-policy data..."*
>
> We agree these are important. Our paper focuses on **OR optimization modeling**: we study **localized corrective adaptation for OR tasks**. Interventions are sparse (~2.6%), teacher swap yields similar performance, and distillation underperforms localized correction. A general on-policy/off-policy investigation is valuable future work.

---

> > ### Author Rebuttal · Reviewer_gpH7 · 2026-04-01
> >
> > Thanks for the response.
> >
> > To make the experiments and results more convincing, could you please detail the data and training configurations and hyperparameters for each experiment run, including how the data for each run is obtained?

---

> > > ### Author Response · Authors · 2026-04-01
> > >
> > > Update (Apr 7, AOE; edited from Apr 1 reply)
> > >
> > > As the discussion period approaches its end in AOE time, we just wanted to briefly ask whether any further specific clarification on the added runs would be helpful. Our Apr 1 follow-up above was intended to address the requested run-level data/training configurations, hyperparameters, and per-run data provenance in a matched-setting format.
> > >
> > > If there are any particular hyperparameters, configuration details, or provenance details that would still be helpful to specify more explicitly, we would be very happy to clarify them while the discussion window remains open.
> > >
> > > Thank you again for your time and consideration.
> > >
> > > ---
> > >
> > > Thank you for this very helpful follow-up. This is exactly the right question for the added rebuttal controls, and it helped us present the comparison more cleanly. The added runs are informative only if the run-level provenance is explicit and the matching is easy to verify, so we summarize the core comparison below in the most direct audit form.
> > >
> > > In the core CALM-vs-distillation comparison, **the only intended change is the source of the SFT training trajectory**. We keep the student, the CALM SFT set, the tool-enabled scaffold, the SFT/RL recipe, and the evaluation protocol fixed, so that the added controls isolate exactly the mechanism of interest: **localized correction of student-native trajectories vs. full teacher-trace replacement**.
> > >
> > > ## 1. Core matched comparison: what changes, and what happens?
> > >
> > > All three runs below use the **same 112-example CALM SFT set**: this is the final CALM SFT set shown in Fig. 6, obtained from the 549-problem SFT pool under the CALM filtering procedure described in Sec. 3.2 / App. E.3. For the two distillation runs, we keep this same SFT set and replace only the training trajectory with a full teacher-generated trace under the same scaffold.
> > >
> > > | Run | SFT training trajectory used on the same 112-example CALM SFT set | Macro Avg on the 3 hardest benchmarks |
> > > |---|---|---:|
> > > | CALM + RL (STORM) | `CALM-corrected student-native trajectories` | 54.9 |
> > > | DeepSeek-R1 distill + RL | `Full teacher-generated trajectories from DeepSeek-R1-0528` under the same scaffold | 47.4 |
> > > | Gemini-2.5-Pro distill + RL | `Full teacher-generated trajectories from Gemini-2.5-Pro` under the same scaffold | 30.8 |
> > >
> > > This is the direct mechanism comparison we intended the added rebuttal runs to answer: **same student, same CALM SFT set, same train size, same downstream RL, same evaluation; under a matched setup, localized correction outperforms full teacher-trace replacement.**
> > >
> > > ## 2. What is held fixed across these three runs?
> > >
> > > | Shared setting | Value |
> > > |---|---|
> > > | Base student / initialization | Qwen3-4B-Thinking-2507, same initialization |
> > > | SFT set | Same 112-example CALM SFT set described above |
> > > | SFT train size | 112 trajectories for each run |
> > > | Prompt template / scaffold | Same prompt template and same tool-enabled scaffold (code interpreter, max 4 executions per turn) |
> > > | SFT hyperparameters | Same as Table 5 / App. E.3 |
> > > | RL initialization rule | RL starts from the final SFT checkpoint of the corresponding run |
> > > | RL training data | Same RL-pool partition (Table 4) |
> > > | RL objective / reward | Same GRPO training and same binary correctness reward (Eq. 3) |
> > > | RL hyperparameters | Same as Table 6 / App. E.3–E.4 |
> > > | Evaluation | Same 8-sample pass@1 protocol (temp=0.6, top-p=0.95), same max 4 code executions |
> > > | Hardware | Same training environment |
> > >
> > > To avoid introducing new confounders, we intentionally kept the data pipeline fixed for the distillation runs and changed only the training trajectory.
> > >
> > > ## 3. Intervener swap within CALM curation
> > >
> > > As a separate robustness check on intervener dependence within the CALM curation stage, we also **swapped only the intervener** while keeping the raw 549-problem curation pool, the intervention budget (`N = 5`), the CALM filtering rule, and all downstream SFT/RL/evaluation settings fixed.
> > >
> > > | Dimension | **Gemini-2.5-Pro** as intervener | **DeepSeek-R1-0528** as intervener |
> > > |---|---|---:|
> > > | Reasoner | Qwen3-4B-Thinking-2507 | same |
> > > | Raw curation pool | same 549 problems | same |
> > > | CALM curation rule | identical | identical |
> > > | Retained golden trajectories | 112 | 55 |
> > > | Downstream SFT/RL/eval | same recipe | same recipe |
> > > | Final Macro Avg | 68.9 | 67.8 |
> > >
> > > We treat this intervener-swap result as a robustness check on CALM curation, not as the matched-count mechanism comparison above. Its role is narrower: under the same CALM procedure, changing the intervener changes the retained set size, while the final downstream performance remains similar within the tested range.
> > >
> > > We will add this run-level configuration summary, together with the exact retained-data counts and per-run settings for the rebuttal-added experiments, to the revised appendix for full transparency.

---

### Official Review · Reviewer_1Qda · 2026-03-13

**Soundness:** 3
**Presentation:** 4
**Significance:** 2
**Originality:** 2
**Overall Recommendation:** 4
**Confidence:** 3

**Summary:**

This paper proposes CALM, a corrective adaptation framework for large reasoning models on optimization modeling tasks. The method uses lightweight corrective hints to refine flawed reasoning trajectories, followed by supervised fine-tuning and reinforcement learning. Based on this framework, the authors build STORM, a 4B model that achieves strong average performance across five optimization modeling benchmarks.

**Compliance With Llm Reviewing Policy:**

Affirmed.

**Key Questions For Authors:**

1. Can you further isolate the effect of CALM-style data correction from the effect of RL?
2. How expensive is the intervention pipeline in practice, and how sensitive is performance to hint quality?
3. Have you compared against stronger baselines such as self-correction, trajectory rewriting, or teacher-generated reasoning traces?

**Limitations:**

The paper discusses some technical limitations indirectly through its analysis and setup, but the discussion of limitations and potential negative societal impact could be more explicit. In particular, the authors could better address:

- the dependence on a strong external intervener model, including cost and reproducibility;
- possible benchmark over-specialization rather than broader generalization;
- risks of deploying incorrect optimization models in real decision-making settings, where errors could have operational or economic consequences.

A short, explicit limitations section covering these points would improve the paper.

**Strengths And Weaknesses:**

Strengths
1. Studies an important and practical task at the intersection of reasoning, code generation, and optimization.
2. The motivation is clear: standard non-reflective supervision may be mismatched with modern reasoning models.
3. The proposed hint-based correction mechanism is intuitive and reasonably novel.
4. Empirical results are strong, especially given the small model size.

Weaknesses
1. The contribution of CALM itself versus the downstream RL stage is not fully isolated.
2. The reliance on an expert intervener raises questions about scalability and cost.
3. Comparisons to stronger alternative adaptation baselines could be more comprehensive.
4. More evidence on robustness and out-of-distribution generalization would strengthen the paper.

---

> ### Author Rebuttal · Authors · 2026-03-29
>
> We thank the reviewer for the careful reading and constructive questions. Below we address the main concerns directly.
>
> Below, we will use **W1** to refer to **Weakness 1**, **Q1** for **Question 1**, and so on.
>
> ## Concern 1 (W1, Q1, part of W3/Q3): What does CALM add beyond RL and teacher-trace replacement?
>
> We agree that this attribution should be explicit. Our claim is not that CALM-SFT must be the strongest SFT checkpoint in isolation. The narrower claim is that CALM-style localized correction provides a better starting policy for downstream RL.
>
> Figure 7a already isolates this point. The two runs start from the same base model, use the same RL algorithm and budget, and differ only in the SFT data source: CALM-corrected trajectories vs. unguided trajectories. Under this matched setup, RL with CALM stays above RL without CALM throughout the observed training window. This supports better RL learning dynamics, not just a one-time gain in starting score.
>
> We also tested the reviewer's stronger replacement-style alternative, teacher-generated reasoning traces, with the same student and the same downstream RL stage:
>
> | Pipeline | MAMO-Complex | IndustryOR | OptMath | Avg |
> |---|---:|---:|---:|---:|
> | CALM + RL | 70.3 | 50.0 | 44.5 | 54.9 |
> | DeepSeek-R1 distillation + RL | 62.5 | 50.4 | 29.5 | 47.4 |
> | Gemini-2.5-Pro distillation + RL | 44.1 | 28.3 | 20.0 | 30.8 |
>
> A supplementary matched-budget checkpoint view shows the same pattern:
>
> | Step | CALM + RL | R1 distill + RL | Gemini distill + RL |
> |---:|---:|---:|---:|
> | 0 | 0.455 | 0.464 | 0.273 |
> | 20 | 0.477 | 0.464 | 0.293 |
> | 40 | 0.493 | 0.466 | 0.299 |
> | 60 | 0.489 | 0.468 | 0.318 |
> | 80 | 0.503 | 0.472 | 0.319 |
>
> So the effect is not explained by RL alone, nor by full teacher-trace replacement + RL. Full teacher-trace supervision is already a stronger replacement-based control than trajectory rewriting because it replaces the entire student trajectory. Since even this stronger control underperforms CALM, the value of localized correction is directly supported.
>
> For Q3's self-correction baseline, we ran OptiMUS (a representative agent-based self-correction method) on the same backbone:
>
> | Method | Macro Avg |
> |---|---:|
> | Qwen3-4B-Thinking-2507 (base) | 57.1 |
> | OptiMUS (Qwen3-4B-Thinking-2507) | 60.4 |
> | STORM | **68.9** |
>
> Generic self-correction without execution-grounded localized intervention does not close the gap.
>
> ## Concern 2 (W2, Q2): Lightweight curation and robustness to teacher choice
>
> We directly tested teacher dependence by swapping Gemini-2.5-Pro for open-source DeepSeek-R1-0528 only during the CALM curation stage, while keeping filtering, SFT, and RL fixed. The table below therefore reports final student performance after the same CALM + RL pipeline:
>
> | Intervener used during CALM curation | Final student avg after CALM + RL | Lift |
> |---|---:|---:|
> | Gemini-2.5-Pro | 68.9 | +11.8 |
> | DeepSeek-R1-0528 | 67.8 | +10.7 |
>
> The final gap is only 1.1 points.
>
> We also summarize teacher-side curation quality and the estimated curation token budget. We report token ranges rather than dollar cost:
>
> | Intervener | Intervener ratio | Hinting Accuracy | Fix rate on initially wrong problems | Estimated input toks | Estimated output toks |
> |---|---:|---:|---:|---:|---:|
> | Gemini-2.5-Pro | ~2.6% | 80.69% | 49.74% | 4.12M-7.13M | 0.67M-0.86M |
> | DeepSeek-R1-0528 | ~2.8% | 75.23% | 29.03% | 2.54M-5.19M | 0.49M-0.63M |
>
> The input-token column should be read only as a rough prompt-budget proxy under the same curation procedure; the clearer difference is on the output side, where Gemini produces somewhat longer intervention responses. More importantly, despite these teacher-side differences, the final student remains close after the same downstream pipeline. In both settings the teacher modifies only ~2.6% / ~2.8% of the student trajectory, so CALM is better understood as localized process-level correction, not full teacher replacement.
>
> ## Concern 3 (W4): Beyond one teacher, one student, and one benchmark suite
>
> Our claim here is still narrow: we do not claim universal transfer. The evidence we provide is that the effect is not tied only to one proprietary teacher, one 4B student, or the current OR test suite.
>
> Together with the teacher-swap result above, we additionally ran the same pipeline on a different student family:
>
> | Student | Base | + SFT | + RL |
> |---|---:|---:|---:|
> | DeepSeek-R1-0528-Qwen3-8B | 58.4 | 60.2 | 67.8 |
>
> We also added an out-of-domain math experiment:
>
> | Benchmark | RL without CALM | RL with CALM |
> |---|---:|---:|
> | aime25 | 68.33 | 69.58 |
> | amc23 | 91.71 | 94.12 |
> | math500-level5 | 81.06 | 83.30 |
> | Avg | 80.37 | 82.33 |
>
> These results suggest that CALM is not only teaching benchmark-specific optimization templates; it improves a more transferable behavior of learning from localized correction and using external computation more effectively.

---

### Official Review · Reviewer_2xx6 · 2026-03-14

**Soundness:** 2
**Presentation:** 2
**Significance:** 2
**Originality:** 2
**Overall Recommendation:** 4
**Confidence:** 3

**Summary:**

This paper introduces CALM to enhance LRMs for optimization modeling by leveraging their native, multi-turn reasoning capabilities. This paper first identify two primary flaws in unguided LRMs: code utilization distrust and lack of OR expertise. Then they propose to address these two flaws by injecting solver grounded hints (through an intervener based on a powerful LLM) to correct specific reasoning flaws in LRMs while preserving their native reflective patterns. After getting the refined trajectories, they use them for SFT and RL on Qwen3-4B. Experiments show that the proposed method can achieve SOTA results on five OR benchmarks, outperforming the DS-R1-0528 model.

**Compliance With Llm Reviewing Policy:**

Affirmed.

**Final Justification:**

I think the authors have addressed most of my concerns. I would like to raise my rating. Please include the results during the rebuttal into the camera-ready version of the paper.

**Key Questions For Authors:**

1. How sensitive are results to the Intervener’s capability?
2. Can you disaggregate improvements by error category after each stage (SFT, RL) with precise rates and provide examples that illustrate systematic reduction of specific triggers?
3. What is the executability rate of generated code and the distribution of solver statuses across datasets and stages?
4. How portable is STORM across solver backends (e.g., Pyomo, OR-Tools, Gurobi) and problem classes beyond the current benchmarks? Any OOD tests?

**Limitations:**

Yes

**Strengths And Weaknesses:**

Strengths:
1. Presentation: This paper is well-organized. The idea is simple and easy to follow.
2. Motivation: I like the section 2.3: A Taxonomy of Flaws in LRM’s Native Reasoning. The analysis makes the proposed idea well-motivated.
3. Experiment: The authors evaluate the proposed model across five benchmarks (NL4OPT, MAMO-Easy, MAMO-Complex, IndustryOR, OptMath) and compare it to a set of baselines (LLMs and LRMs with much more parameters, agent-based methods, and learning-based methods). The proposed model is able to outperform the DS-R1-0528 (671B) model.

Weaknesses:
1. Method: It seems to me that the proposed data synthesis method is heavily reliant on a closed-source, much stronger model (Gemini-2.5-Pro) as the Intervener. I am not sure if the proposed method also works when we use an open-source, weaker model. It is also unknown whether the performance improvements come from the indirect "distillation" of a stronger model. It would be interesting to test Gemini-2.5-Pro on these 5 benchmarks and a baseline that directly uses the Gemini as the teacher model to cold start (SFT) then RL.
2. More ablation studies would be needed. For example, different combinations of the reasoner (only 4B model is used, not sure how the behaviors differs when we increase the size of the model, does the proposed method still can bring significant improvements?) and the intervener, number/type of hints, ablations to solver access (no code execution), or alternative reflective training.
3. The pilot study indicating “non-reflective SFT hurts” is useful but would be stronger if accompanied by a matched “reflective-SFT” baseline (e.g., SFT on unguided reflective traces) and a combination SFT of reflective + non-reflective to isolate the proposed mechanism.
4. The proposed idea is not very exciting. Intergraing execution feedback into the reasoning loop and guide the corrections is widely used for a wide range of the domains. Also, it is well-known that reflective patterns helps for complex reasoning tasks.
5. Typo: The caption of figure 4 says the method is called "CLAM".
6. Related work is short and is placed in the Appendix. A more thorough discussion of existing works is needed.

---

> ### Author Rebuttal · Authors · 2026-03-28
>
> We sincerely thank the reviewer for the careful reading and thoughtful suggestions.
>
> Below, we use **W1** for **Weakness 1**, **Q1** for **Question 1**, etc.
>
> ## W1, Q1: Intervener dependence and indirect distillation
>
> We tested teacher sensitivity and the suggested distillation baseline on the same Qwen3-4B-Thinking-2507 student.
>
> | Setting | Final Macro Avg | Lift |
> |---|---:|---:|
> | Gemini-2.5-Pro intervener | 68.9 | +11.8 |
> | DeepSeek-R1-0528 intervener | 67.8 | +10.7 |
>
> The gap is only **1.1 pt**, showing the gain is not tied to one proprietary intervener.
>
> We also tested **direct distillation** (SFT on full teacher traces), then the same RL:
>
> | Pipeline | MAMO-Complex | IndustryOR | OptMath | Avg |
> |---|---:|---:|---:|---:|
> | CALM + RL (=STORM) | **70.3** | 50.0 | **44.5** | **54.9** |
> | DeepSeek-R1 distill + RL | 62.5 | **50.4** | 29.5 | 47.4 |
> | Gemini distill + RL | 44.1 | 28.3 | 20.0 | 30.8 |
>
> The gain is not explained by indirect distillation. CALM modifies <2.6% of tokens, preserving >97.4% of the student trajectory via localized correction rather than full-trace replacement.
>
> ## W2: Additional ablations
>
> **Cross-student generalization:**
>
> | Student | Base | +SFT | +RL |
> |---|---:|---:|---:|
> | DeepSeek-R1-0528-Qwen3-8B | 58.4 | 60.2 | 67.8 |
>
> The method generalizes beyond one 4B student. Fuller hint-type ablations and cross-backend portability are important follow-up directions. For alternative reflective training, please see W3 below.
>
> ## W3: Reflective-SFT baseline
>
> We compared several SFT data sources on the same student:
>
>
> | SFT data source | NL4OPT | MAMO-Easy | MAMO-Complex | IndustryOR | OptMath | Macro Avg |
> |---|---:|---:|---:|---:|---:|---:|
> | Qwen3-4B-Thinking-2507 (base) | 85.8 | 73.8 | 46.5 | 46.2 | 33.1 | 57.1 |
> | OR-Instruct-3K | 92.9 | 88.7 | 40.5 | 27.5 | 6.6 | 51.2 |
> | Reflective OR-Instruct | 85.4 | 78.0 | 38.5 | 33.5 | 24.6 | 52.0 |
> | CALM SFT | 86.6 | 77.9 | 54.3 | 44.1 | 30.4 | 58.7 |
>
> (i) Non-reflective SFT helps easy sets but collapses on harder ones; (ii) generic reflective SFT remains weak (52.0 avg) vs. CALM SFT (58.7). This isolates the value of CALM-specific localized correction.
>
> ## W4: Novelty
>
> We do **not** claim execution feedback in reasoning loops is novel generally. Our contribution is more specific:
>
> 1. **OR adaptation finding**: Most standard strategies (non-reflective SFT, generic reflective SFT, full distillation) fail on complex OR benchmarks. This negative result reveals that OR's unique solver-interaction demands are not well-served by off-the-shelf methods.
>
> 2. **Localized correction**: Unlike prior methods that replace the entire trajectory, CALM preserves >97.4% of student tokens and injects corrections only at flaw points.
>
> 3. **Empirical validation**: Across all five benchmarks, CALM SFT (58.7) > reflective SFT (52.0) > non-reflective SFT (51.2); on the three hard benchmarks (MAMO-Complex / IndustryOR / OptMath), CALM+RL (54.9) >> distillation+RL (47.4 / 30.8).
>
> ## Q2: Stage-wise improvement by error category
>
> | Error category | Base | SFT | RL | Base -> SFT | SFT -> RL | Base -> RL |
> |---|---:|---:|---:|---:|---:|---:|
> | Code Utilization Distrust | 1.00 | 0.86 | 0.60 | -14.4% | **-30.1%** | -40.2% |
> | Lack of OR Expertise | 1.00 | 0.68 | 0.48 | **-32.3%** | -28.6% | -51.6% |
>
> SFT mainly reduces OR-expertise flaws; RL further reduces code-utilization distrust. Figure 11/Appendix I shows this qualitatively.
>
> ## Q3: Code executability and solver-status distribution
>
> We report the executability rate and solver-status distribution across stages:
>
> | Stage | Code use | Exec success | `Optimal` status | Timeout |
> |---|---:|---:|---:|---:|
> | Base | 92.2% | 91.4% | 42.1% | 20.2% |
> | SFT | 89.4% | 92.9% | 55.1% | 22.2% |
> | RL | **95.5%** | **96.7%** | **62.3%** | **14.3%** |
>
> RL both increases code utilization and makes solver interaction more reliable, with Optimal rate rising from 42.1% to 62.3% while Timeout drops from 20.2% to 14.3%.
>
> ## Q4: OOD generalization
>
> | OOD Benchmark | RL w/o CALM | RL w/ CALM |
> |---|---:|---:|
> | aime25 | 68.33 | **69.58** |
> | amc23 | 91.71 | **94.12** |
> | math500-level5 | 81.06 | **83.30** |
> | Average | 80.37 | **82.33** |
>
> CALM teaches transferable tool-augmented reasoning beyond the current OR test sets. Regarding solver backends, STORM generates standard Python optimization code (via PuLP/Gurobi) rather than solver-specific APIs, so switching backends requires no model change.
>
> ## Minor issues
>
> We will fix the Figure 4 typo ("CLAM") and expand the related-work section to cover self-refinement/self-correction in LLMs, knowledge distillation for reasoning, and RL-based code generation.

---

> > ### Author Rebuttal · Reviewer_2xx6 · 2026-04-01
> >
> > I think the authors have addressed most of my concerns. I would like to raise my rating. Please include the results during the rebuttal into the camera-ready version of the paper.

---

> > > ### Author Response · Authors · 2026-04-01
> > >
> > > Thank you again for the careful review and for your **updated assessment** after the additional controls.
> > >
> > > We especially appreciate that, from the start, you recognized the value of the flaw taxonomy, the motivation in Sec. 2.3, and the breadth of the benchmark evaluation. Your follow-up requests then pushed us to make the paper substantially more rigorous.
> > >
> > > In direct response to your concerns, we added the open-source intervener swap, the matched teacher-trace distillation baselines, the reflective-SFT comparison, the stage-wise error-category breakdown, the executability / solver-status analysis, and the OOD / cross-student checks. These additions significantly sharpened the final claim and make the evidence much cleaner.
> > >
> > > In particular, the revised evidence now supports a much more precise conclusion: the gain is not explained by one proprietary intervener, by generic reflective supervision, or by full teacher-trace replacement, but by **localized correction of the student's own executable reasoning trajectories**.
> > >
> > > We are also grateful for the concrete presentation comments; we will fix the Figure 4 typo and expand the related-work discussion in the revision. Thank you again — your feedback **materially strengthened the paper**.

---

### Decision · Program_Chairs · 2026-04-30

**Decision:**

Accept (regular)

**Comment:**

This paper proposes CALM, a corrective adaptation framework for large reasoning models (LRMs) applied to optimization modeling tasks. The method introduces a reasoner–intervener paradigm, where model-generated reasoning trajectories are minimally corrected through targeted hints, followed by supervised fine-tuning and reinforcement learning (STORM). Empirical results on several operations research benchmarks demonstrate strong performance, in some cases approaching or matching significantly larger models.

The reviewers raised several concerns. First, the contribution of CALM itself is not clearly disentangled from the effects of the downstream RL stage, making it difficult to assess the independent impact of each component. Second, the reliance on an expert intervener introduces potential issues of scalability and cost. Third, the comparisons to stronger alternative adaptation baselines are not sufficiently comprehensive. Finally, additional evidence on robustness and out-of-distribution generalization would further strengthen the paper.

Overall, while the idea of corrective adaptation is interesting, the current submission falls short in isolating its core contributions, demonstrating scalability, and providing sufficiently rigorous evaluation.